# ACTIVE LEARNING FOR CONTINUAL LEARNING: KEEPING THE PAST ALIVE IN THE PRESENT

**Jaehyun Park[1], Dongmin Park[2], Jae-Gil Lee[1]***
[1] KAIST, [2] KRAFTON
{jhpark813, jaegil}@kaist.ac.kr, dongmin.park@krafton.com

## ABSTRACT

*Continual learning (CL)* enables deep neural networks to adapt to ever-changing data distributions. In practice, there may be scenarios where annotation is costly, leading to *active continual learning (ACL)*, which performs *active learning (AL)* for the CL scenarios when reducing the labeling cost by selecting the most informative subset is preferable. However, conventional AL strategies are not suitable for ACL, as they focus solely on learning the new knowledge, leading to *catastrophic forgetting* of previously learned tasks. Therefore, ACL requires a new AL strategy that can balance the prevention of catastrophic forgetting and the ability to quickly learn new tasks. In this paper, we propose **AccuACL**, Accumulated informativeness-based Active Continual Learning, by the novel use of the Fisher information matrix as a criterion for sample selection, derived from a theoretical analysis of the Fisher-optimality preservation properties within the framework of ACL, while also addressing the scalability issue of Fisher information-based AL. Extensive experiments demonstrate that AccuACL significantly outperforms AL baselines across various CL algorithms, increasing the average accuracy and forgetting by 23.8% and 17.0%, respectively, on average.

## 1 INTRODUCTION

*Continual learning (CL)*, a learning scenario to adapt models continuously on evolving data distributions, is essential in our dynamic world (Thrun, 1995). Numerous CL methods have been advanced with the common goal of preserving past knowledge while acquiring new knowledge across the CL tasks (Abraham and Robins, 2005; Kim et al., 2023b; Mermillod et al., 2013). While most studies in CL assume that the evolving data distributions are fully labeled, this is rarely the case in practice. For example, fraud detection systems in financial services must continuously learn to recognize new fraud patterns. However, the process of annotating these unique patterns is expensive, since it requires professional analysis in the field (Lebichot et al., 2024). As a result, *active continual learning (ACL)* is becoming a key challenge for effectively mitigating the limited labeling budget, by querying the most important examples at each CL task that maximize the model's performance over *all* observed tasks (Cai et al., 2022; Perkonigg et al., 2021; Vu et al., 2023).

However, conventional *active learning (AL)* strategies are *not* suitable for ACL scenarios, because they are mainly designed to query the examples relevant to the knowledge about a new task. That is, in both uncertainty-based and diversity-based AL strategies, the examples that the model has not encountered are highly prioritized (Ash et al., 2019; Sener and Savarese, 2017; Settles, 1995). Figure 1(a) first illustrates the inappropriateness of the conventional AL strategies for ACL scenarios. The unlabeled data with diverse feature importance is continuously received for each task. At task $t$, these AL strategies typically pay more attention to new features (i.e., Features 3 and 4), and accordingly, the examples mainly involved with the new features are selected by the active learner. That is, the AL strategies focus only on quickly learning new tasks and neglect preventing catastrophic forgetting. Thus, the past knowledge involved with Features 1 and 2 can be forgotten after learning task $t$. Failing to capture the crucial features of the past knowledge causes *catastrophic forgetting* (French, 1999), which results in significant performance degradation even compared to random querying, as shown in Figure 1(b).

---

*Corresponding author.

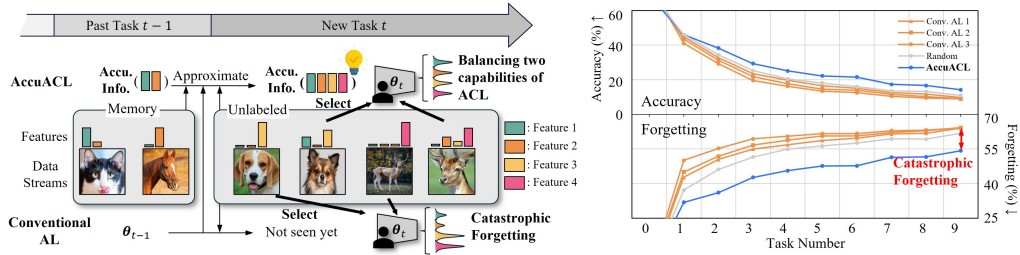

(a) Accumulated informativeness in ACL.    (b) AL performance on CL benchmarks.

Figure 1: Overview of AccuACL; (a) Unlike conventional AL strategies that only focus on the new task and cause catastrophic forgetting, AccuACL balances the prevention of catastrophic forgetting and the ability to learn new tasks quickly, by defining the *accumulated informativeness*; (b) shows the catastrophic forgetting of the conventional AL strategies on SplitCIFAR100.

This paper offers a novel viewpoint on the query strategies in ACL. Combined with the evolving data distributions of CL, it is very important not to forget the knowledge learned from past tasks. As an example with Figure 1(a), all of the four features are important in ACL because they have appeared at one of the past or new tasks. However, the data for past tasks is inherently *not* accessible in CL, which makes it challenging to identify the important features from past tasks at the new task. In summary, the key technical challenge in the paper lies in answering the question "what examples in the new task contribute to the preservation of past knowledge?"

To answer the aforementioned question, we formulate the *accumulated informativeness* as a novel standard for informativeness in ACL, which balances the prevention of catastrophic forgetting and the ability to quickly learn new tasks. It enables assessing an example's informativeness with respect to both past and new tasks. Then, we introduce **AccuACL** (Accumulated informativeness-based Active Continual Learning), an algorithm based on the theoretical analysis of the *Fisher information-based AL* (Sourati et al., 2017; Zhang and Oles, 2000), that maximizes the accumulated informativeness. We model the accumulated informativeness via the Fisher information matrix, through the approximation with a small memory buffer commonly adopted by many rehearsal-based CL methods, the model parameters, and the unlabeled data pool for the new task, as illustrated in Figure 1(a). As a result, AccuACL becomes to prefer the examples that comprehensively contain the four important features (i.e., Features 1∼4) in Figure 1(a). Furthermore, because Fisher information-based AL is a combinatorial optimization problem, we develop a query algorithm based on two theoretical properties that an optimal labeled subset should possess. To the best of our knowledge, this is the first ACL study that offers the ability to avoid catastrophic forgetting.

Extensive experiments with four different CL methods on three CL benchmarks, SplitCIFAR10, SplitCIFAR100, and SplitTinyImageNet, show that AccuACL significantly boosts CL approaches, outperforming conventional AL baselines by 23.8% and 17.0%, in terms of average accuracy and forgetting, respectively, on average. As shown in Figure 1(b), AccuACL always achieves dominance in both metrics throughout the entire sequence of tasks. The source code is available at https://github.com/kaist-dmlab/AccuACL.

## 2 RELATED WORK

**Active Learning (AL).** AL is a research field that focuses on the selection of unlabeled data points for labeling by an oracle, which provides supervision, especially in domains where labeling requires significant costs (Ren et al., 2021; Tharwat and Schenck, 2023). This process allows us to optimize model performance under labeling budget constraints. Many strategies exist for selecting informative examples from unlabeled data. Uncertainty-based approaches measure model prediction uncertainty and select the most uncertain examples (Roth and Small, 2006; Settles, 1995; Wang and Shang, 2014). Diversity-based methods prioritize diverse examples to reduce redundancy between selected examples and capture a broader range of patterns and complexities in the data pool (Sener and Savarese, 2017). Hybrid approaches use gradient space embedding to select diverse and uncertain examples (Ash et al., 2019). Fisher information-based methods measure the asymptotic value of unlabeled data with theoretical guarantee (Sourati et al., 2017; Zhang and Oles, 2000). Notably, Kirsch and Gal (2022) demonstrate that modern AL algorithms optimize the same Fisher-based objective.

**Continual Learning (CL).** CL has gained interest as a way to adapt models to new tasks over time. Numerous research has investigated different methods to address the stability-plasticity dilemma (Mermillod et al., 2013). Rehearsal-based methods store (Aljundi et al., 2019; Buzzega et al., 2020; Caccia et al., 2022; Rahaf and Lucas, 2019; Rolnick et al., 2019; Liang and Li, 2024; Kim et al., 2023a; Lin et al., 2024) or generate (Shin et al., 2017) a subset of examples from past tasks at a constant cost. These examples are then replayed for the new task to retain past knowledge. Regularization-based methods discourage significant changes to model parameters that are essential for past tasks (Aljundi et al., 2018; Kirkpatrick et al., 2017). Dynamic-structure-based methods generate distinct modules to augment the ability to learn new tasks (Yan et al., 2021; Zhou et al., 2023; Wang et al., 2023). Rehearsal-free or prompt-based methods are gaining popularity in the CL field, using pre-trained models and prompt tuning to adjust to data distribution shifts (Wang et al., 2022a;b).

**Active Continual Learning (ACL).** There has been a lack of research effort on ACL. Vu et al. (2023) support our goal of tackling the impracticality of fully-supervised datasets in real-world settings; however, instead of developing their own AL approach for CL, they recommend a combination of AL and CL strategies for various CL scenarios. Ayub and Fendley (2022) discuss ACL in a few-shot situation; however, they choose examples based on their proposed uncertainty measure, which focuses on learning new tasks. Perkonigg et al. (2021) address the need for ACL in a medical domain; however, because they assume that the distribution is gradually changing, their focus is on detecting the change in the distribution to determine when to label incoming images.

## 3 PRELIMINARY

### 3.1 PROBLEM SETUP: ACTIVE CONTINUAL LEARNING

Consider a sequence of tasks $\mathcal{T} = \{1, \ldots, T\}$, where the input space $\mathcal{X}_t$ and label space $\mathcal{Y}_t$ shift as the tasks progress. This study focuses on *class-incremental* learning, where $\mathcal{Y}_t \cap \mathcal{Y}_{t'} = \emptyset$ ($t' < t$). In the *active continual learning (ACL)* framework, at task $t$, an unlabeled dataset $U_t = \{\mathbf{x}_t^i\}_{i=1}^{n_t} \sim D_{\mathcal{X}_t}$ is provided. Within a labeling budget constraint $b_t$, we query the label of the selected subset $S_t \subset U_t$ to the oracle labeler $\mathcal{A}(\cdot)$ to obtain $L_t = \{(\mathbf{x}_t^i, y_t^i)\}_{i=1}^{|S_t|}$, where $y_t^i \sim D_{\mathcal{Y}_t|\mathcal{X}_t}(\mathbf{x}_t^i)$. The objective of ACL is to identify the most informative subset $S_t^*$ such that, when the parameter $\hat{\boldsymbol{\theta}}_t$ is trained by $S_t^*$, it minimizes an expected arbitrary error $\epsilon$ across *all* encountered data distribution $D_{1:t} = D_{\mathcal{X}_{1:t} \times \mathcal{Y}_{1:t}}$. Formally, at each task $t$,

$$S_t^* = \underset{S_t \subset U_t, |S_t| \leq b_t}{\arg\min} \mathbb{E}_{(\mathbf{x}, y) \sim D_{1:t}} \big[\epsilon(\mathbf{x}, y; \hat{\boldsymbol{\theta}}_t)\big] \ s.t. \ \hat{\boldsymbol{\theta}}_t = \underset{\boldsymbol{\theta}}{\arg\min} \mathcal{L}_{\text{CL}}(\boldsymbol{\theta}; \boldsymbol{\theta}_{t-1}, \mathcal{A}(S_t)), \quad (1)$$

where $\epsilon$ can be the cross-entropy error for classification tasks. Note that AL for each CL task $t \in \mathcal{T}$ is performed through multiple rounds, as in conventional AL.

### 3.2 FISHER INFORMATION-BASED ACTIVE LEARNING

The *Fisher information matrix* is often used to quantify the information conveyed to the model parameters, indicating their significance in modeling a data distribution (Lehmann and Casella, 2006). A model parameter $\theta \in \boldsymbol{\theta}$ with a high Fisher information is essential for modeling the distribution, and altering its value hinders the model performance. Fisher information-based AL assumes the true model parameter $\boldsymbol{\theta}^*$, in which the underlying conditional distribution $D_{\mathcal{Y}|\mathcal{X}}(\mathbf{x}) = p(\cdot|\mathbf{x}, \boldsymbol{\theta}^*)$. Under this premise, Fisher information-based AL seeks to select an optimal subset $S^* \subset U$ to label, which minimizes the discrepancy between the trained model parameter $\hat{\boldsymbol{\theta}}$ and the true model parameter $\boldsymbol{\theta}^*$. In particular, when the discrepancy is formulated as the log-likelihood ratio $\epsilon(\mathbf{x}, y, \hat{\boldsymbol{\theta}}, \boldsymbol{\theta}^*) = \log p(y|\mathbf{x}; \hat{\boldsymbol{\theta}}) - \log p(y|\mathbf{x}; \boldsymbol{\theta}^*)$, it results in a simpler objective function that leverages the Fisher information matrices (Zhang and Oles, 2000). Formally,

$$S^* = \underset{S \subset U, |S| \leq b}{\arg\min} \mathbb{E}_{(\mathbf{x}, y) \sim D} \big[\epsilon(\mathbf{x}, y; \hat{\boldsymbol{\theta}})\big] \ s.t. \ \hat{\boldsymbol{\theta}} = \underset{\boldsymbol{\theta}}{\arg\min} \mathcal{L}(\boldsymbol{\theta}; \mathcal{A}(S)) \quad (2)$$

$$= \underset{S \subset U, |S| \leq b}{\arg\min} \mathbb{E}_{\mathbf{x} \sim U} \big[\mathbb{E}_{y \sim D_{\mathcal{Y}|\mathcal{X}}(\mathbf{x})} \big[\epsilon(\mathbf{x}, y, \hat{\boldsymbol{\theta}}, \boldsymbol{\theta}^*)\big]\big] \ s.t. \ \hat{\boldsymbol{\theta}} = \underset{\boldsymbol{\theta}}{\arg\min} \mathcal{L}(\boldsymbol{\theta}; \mathcal{A}(S)) \quad (3)$$

$$= \underset{S \subset U, |S| \leq b}{\arg\min} \ \text{tr}\big[\underbrace{\boldsymbol{I}(\boldsymbol{\theta}^*; S)}_{\text{candidate}}{}^{-1} \underbrace{\boldsymbol{I}(\boldsymbol{\theta}^*; U)}_{\text{target}}\big], \quad (4)$$

---

**Algorithm 1** AccuACL

---

INPUT: initial model parameter $\boldsymbol{\theta}_0$, CL tasks $\{1, \ldots, T\}$, unlabeled task data $\{U_1, \ldots, U_T\}$, oracle labeler $\mathcal{A}$, per-round budget $b$, active learning round $R$, memory buffer $M \leftarrow \emptyset$.

1: **for** $t \leftarrow 1$ to $T$ **do**                                                                    ▷ CL Tasks
2:     Querying set $S \leftarrow \texttt{Random}(U_t, b)$                              ▷ Start with initial random querying
3:     Unlabeled data pool $U \leftarrow U_t - S$
4:     $\boldsymbol{\theta}_t \leftarrow \arg\min_{\boldsymbol{\theta}} \mathcal{L}_{\text{CL}}(\boldsymbol{\theta}; \boldsymbol{\theta}_{t-1}, \mathcal{A}(S))$
5:     **for** $r \leftarrow 1$ to $R$ **do**
6:         $\boldsymbol{F}_M \leftarrow \boldsymbol{f}(\boldsymbol{\theta}_t; M), \boldsymbol{F}_U \leftarrow \boldsymbol{f}(\boldsymbol{\theta}_t; U)$          ▷ Get Fisher information *embedding* (§ 4.3)
7:         $F_t \leftarrow \lambda \cdot \overline{\boldsymbol{F}_M} + (1 - \lambda) \cdot \overline{\boldsymbol{F}_U}$                      ▷ Get *target* Fisher information (§ 4.2)
8:         $\mathbf{x} \leftarrow \underset{i \in \{0, \ldots, |U|-1\}}{\arg\max^{(2b)}} \exp\left(-\mathcal{D}_{\text{JS}}(\sigma(F_{U,i})||\sigma(F_t))\right)$     ▷ Over-sample by distribution score (§ 4.5.1)
9:         $\mathbf{x} \leftarrow \arg\max_{i \in \mathbf{x}}^{(b)} \|F_{U,i}\|_2$                              ▷ Sample by magnitude score (§ 4.5.1)
10:        $U \leftarrow U - U[\mathbf{x}], S \leftarrow S \cup U[\mathbf{x}]$
11:        $\boldsymbol{\theta}_t \leftarrow \arg\min_{\boldsymbol{\theta}} \mathcal{L}_{\text{CL}}(\boldsymbol{\theta}; \boldsymbol{\theta}_{t-1}, \mathcal{A}(S))$
12:    $M \leftarrow \texttt{MemoryUpdate}(M, \mathcal{A}(\mathcal{S}))$          ▷ Update the memory buffer with new labeled samples

OUTPUT: $\boldsymbol{\theta}_T$ (final CL model)

---

where $b$ is the labeling budget, $\mathcal{A}(\cdot)$ is the oracle labeler, and $\boldsymbol{I}(\boldsymbol{\theta}; S)$ is the Fisher information matrix over an arbitrary subset $S$, which is equivalent to the covariance matrix of the score function $\nabla_{\boldsymbol{\theta}} \log p(y|\mathbf{x}, \boldsymbol{\theta}) \in \mathbb{R}^{|\boldsymbol{\theta}|}$, formally expressed as

$$\boldsymbol{I}(\boldsymbol{\theta}; S) = -\frac{1}{|S|} \sum_{\mathbf{x} \in S} \sum_{y \in C} p(y|\mathbf{x}; \boldsymbol{\theta}) \nabla_{\boldsymbol{\theta}} \log p(y|\mathbf{x}; \boldsymbol{\theta}) \nabla_{\boldsymbol{\theta}}^T \log p(y|\mathbf{x}; \boldsymbol{\theta}) \in \mathbb{R}^{|\boldsymbol{\theta}| \times |\boldsymbol{\theta}|}. \tag{5}$$

The *target* Fisher information matrix $\boldsymbol{I}(\boldsymbol{\theta}^*; U)$ quantifies the importance of parameters in modeling the distribution $D$ (or the unlabeled data pool $U$), while the *candidate* Fisher information matrix $\boldsymbol{I}(\boldsymbol{\theta}^*; S)$ quantifies the importance of parameters in modeling the candidate subset $S$. Intuitively, training a model using $S^*$ puts emphasis on properly estimating the essential parameter specified by the *target* Fisher information matrix. For the AL scenario, $\boldsymbol{\theta}^*$ is approximated by the estimated parameter from the initially labeled data.

## 4 ACCUACL: PROPOSED ACL METHOD

In this section, we develop a novel ACL strategy that queries data, which prevents catastrophic forgetting and, at the same time, learns new tasks quickly. First, we establish the *accumulated informativeness* to formulate an optimal informativeness measure that accounts for both preventing forgetting and facilitating rapid learning (in § 4.1). Second, we show that the Fisher-based ACL is a promising approach for integrating accumulated informativeness into the query strategy (in § 4.2). Third, we provide an efficient approach for approximating the Fisher information matrix to improve scalability (in § 4.3). Finally, we introduce **AccuACL**, Accumulated informativeness-based Active Continual Learning, which is derived from two unique properties that the approximated Fisher-based ACL should satisfy (in § 4.4). Furthermore, we provide a complexity analysis of AccuACL, demonstrating its practical usability. The overall methodology is provided in Algorithm 1.

### 4.1 ACCUMULATED INFORMATIVENESS

We commence by introducing the accumulated informativeness for ACL, which quantifies the amount to which a subset (or a sample) $S_t \subset U_t$ at task $t$ is beneficial for enhancing the performance within the framework of ACL. Confined to task $t$, the *informativeness* of $S_t$ about an unlabeled dataset $U_t$ is

$$\text{INFO}(S_t; U_t) = \mathbb{E}_{(\mathbf{x}, y) \sim \mathcal{A}(U_t)}\left[p(y|\mathbf{x}; \hat{\boldsymbol{\theta}}_t)\right] \, s.t. \, \hat{\boldsymbol{\theta}}_t = \arg\min_{\boldsymbol{\theta}} \mathcal{L}_{\text{CL}}(\boldsymbol{\theta}; \boldsymbol{\theta}_{t-1}, \mathcal{A}(S_t)), \tag{6}$$

which is the expected likelihood produced by $\hat{\boldsymbol{\theta}}_t$. Here, $\hat{\boldsymbol{\theta}}_t$ is initialized using the parameter $\boldsymbol{\theta}_{t-1}$ at task $t-1$. Conventional AL algorithms, without considering the requirements of CL, work by selecting a subset that maximizes $\text{INFO}(S_t; U_t)$. In the context of ACL, however, it is crucial to consider the influence of the candidate subset $S_t$ on previous tasks, represented by the previous unlabeled data pool

$U_{1:t-1}$. Thus, to achieve a balance between preventing forgetting of past knowledge and facilitating efficient learning of new information, the *accumulated informativeness* $\text{ACCUINFO}(S_t, U_{1:t})$ is defined as a composite of $\text{INFO}(S_t; U_t)$, representing the capacity for rapid acquisition of new tasks, and $\text{INFO}(S_t; U_{1:t-1})$, denoting the capacity to prevent catastrophic forgetting,

$$\text{ACCUINFO}(S_t, U_{1:t}) = f(\text{INFO}(S_t; U_t), \text{INFO}(S_t; U_{1:t-1})), \tag{7}$$

where $f(\cdot)$ can be any function to combine the two informativeness measures.

## 4.2 ACCUMULATED INFORMATIVENESS AND FISHER-BASED ACL

From Eq. (1), we can extend the objective in Eq. (4) to the ACL problem. In contrast to conventional AL, the objective of CL is to maximize the generalization performance across *all* encountered tasks. The Fisher-based AL objective for each task $t$ in ACL is formulated as

$$S_t^* = \underset{S_t \subset U_t, |S_t| \le b_t}{\arg\min} \underbrace{\text{tr}\big[\boldsymbol{I}(\boldsymbol{\theta}_t^*; S_t)^{-1}\boldsymbol{I}(\boldsymbol{\theta}_t^*; U_{1:t})\big]}_{-\text{ACCUINFO}(S_t, U_{1:t})} \tag{8}$$

where $U_{1:t}$ is the whole collection of unlabeled data from all encountered tasks $\{1, \ldots, t\}$, $b_t$ is the labeling budget for task $t$, and $\boldsymbol{\theta}_t^*$ is the true model parameter such that $\boldsymbol{\theta}_t^* = \arg\max_{\boldsymbol{\theta}} \mathbb{E}_{D_{1:t}}[p(y|\mathbf{x}; \boldsymbol{\theta})]$. The *target* Fisher information matrix $\boldsymbol{I}(\boldsymbol{\theta}_t; U_{1:t})$ represents the importance of parameters for all observed tasks. That is, the trace function in Eq. (8) represents the accumulated informativeness in Eq. (7). As a result, the optimal subset $S_t^*$ guarantees an accurate estimate of essential parameters for all observed tasks.

In Theorem 4.1, we show that the target Fisher information matrix in Eq. (8) can be decoupled into two matrices representing past and new information, respectively, with $\lambda$ balancing the informativeness between the two. A number close to 1 indicates an AL that prioritizes preventing catastrophic forgetting, whereas a value close to 0 indicates an AL that prioritizes quick learning of new tasks. That is, Eq. (9) offers a practical form of the function $f(\cdot, \cdot)$ that combines the two informativeness measures for past and new tasks in Eq. (7).

**Theorem 4.1.** *Let $U_{1:t}$ be the unlabeled data pool for all seen tasks until task $t$. Then, the target Fisher information matrix can be divided into past and new information matrices such that*

$$\boldsymbol{I}(\boldsymbol{\theta}_t; U_{1:t}) = \lambda \cdot \underbrace{\boldsymbol{I}(\boldsymbol{\theta}_t; U_{1:t-1})}_{\text{past information}} + (1 - \lambda) \cdot \underbrace{\boldsymbol{I}(\boldsymbol{\theta}_t; U_t)}_{\text{new information}} \tag{9}$$

*with the optimal value of the balancing parameter $\lambda = \frac{|U_{1:t-1}|}{|U_{1:t}|}$.*

*Proof.* The complete proof is available in Appendix A. $\square$

However, it is infeasible to directly apply an existing optimization technique to Fisher-based AL, as developed for conventional AL, to select examples based on Eq. (8) (Ash et al., 2021). The reason is that the unlabeled data pool $U_{1:t-1}$ of the past tasks is *not* available in CL scenarios. Consequently, we leverage a small-sized memory buffer $M_t \subset L_{1:t-1}$, generally maintained in rehearsal-based CL (Aljundi et al., 2019; Buzzega et al., 2020; Rolnick et al., 2019). The *estimated* target Fisher information matrix for task $t$ is defined as

$$\boldsymbol{I}(\boldsymbol{\theta}_t; U_{1:t}) \approx \boldsymbol{I}(\boldsymbol{\theta}_t; M_t, U_t) = \lambda \cdot \boldsymbol{I}(\boldsymbol{\theta}_t; M_t) + (1 - \lambda) \cdot \boldsymbol{I}(\boldsymbol{\theta}_t; U_t), \quad \lambda = \frac{|U_{1:t-1}|}{|U_{1:t}|}. \tag{10}$$

Please refer to Appendix B for further analysis of the estimation.

## 4.3 FISHER INFORMATION EMBEDDING

For deep neural networks with numerous parameters, directly solving Eq. (8) is infeasible owing to the computationally expensive and time-consuming calculation of the inverse of the Fisher information matrix, which is cubic in complexity. Recent research, such as BAIT (Ash et al., 2021), reduces the computational cost by using the last linear classification layer to obtain the Fisher information matrix and an online approach to update the inverse matrix via Woodbury matrix identity. However,

since matrix inversion is still needed, it is computationally demanding, confining its application to small-scale datasets such as MNIST (LeCun et al., 1998) and CIFAR10 (Alex, 2009).

Accordingly, we propose the *Fisher information embedding*, which is the diagonal component of the Fisher information matrix, to reduce spatial and temporal complexity from square to linear. Formally, the *Fisher information embedding* $\boldsymbol{f}(\boldsymbol{\theta}_t; \mathbf{x})$ of an example $\mathbf{x}$ is expressed as

$$\boldsymbol{f}(\boldsymbol{\theta}_t; \mathbf{x}) = \sum_{y \in C} p(y|\mathbf{x}; \boldsymbol{\theta}_t) \left[\nabla_{\boldsymbol{\theta}_t} \log p(y|\mathbf{x}; \boldsymbol{\theta}_t)\right]^2 \in \mathbb{R}^{|\boldsymbol{\theta}_t|}, \tag{11}$$

which is equivalent to the diagonal component of Eq. (5). The use of the diagonal elements of the Fisher information matrix is known to effectively prevent catastrophic forgetting when a model is adapted to shifting data distributions (Kirkpatrick et al., 2017; Pennington and Worah, 2018). While previous studies have used the diagonal Fisher information as a regularization method in gradient descent, we use it as a representation for each example.

Moreover, we consider solely the last linear classification layer to compute the Fisher information matrix, based on the premise that the penultimate layer representation approximates a convex model (Ash et al., 2021). The computation of the Fisher information embedding necessitates $|C|$ backward propagation since the gradient must be derived throughout the whole class space, which may be impractical for large-scale datasets. In Theorem 4.2, we show that its calculation can be reduced into a single forward operation, thereby avoiding the need for heavy computations proportional to $|C|$. This embedding is used to diagonally approximate the Fisher information matrix of a dataset $D$ as $\boldsymbol{F}(\boldsymbol{\theta}_t; D) = \frac{1}{|D|} \sum_{\mathbf{x} \in D} \boldsymbol{f}(\boldsymbol{\theta}_t; \mathbf{x})$. Accordingly, we obtain the embedding version of our target Fisher information $\boldsymbol{F}(\boldsymbol{\theta}_t; M_t, U_t) = \lambda \cdot \boldsymbol{F}(\boldsymbol{\theta}_t; M_t) + (1 - \lambda) \cdot \boldsymbol{F}(\boldsymbol{\theta}_t; U_t)$.

**Theorem 4.2.** *Let $K$ be the number of classes, $d$ be the number of embedding dimensions, $\boldsymbol{\theta}_t^{[L]} = (w_{11}, \ldots, w_{Kd}) \in \mathbb{R}^{Kd}$ be the parameters of the last linear classification layer, and $\mathbf{h}(\boldsymbol{\theta}_t; \mathbf{x}) \in \mathbb{R}^d$ be the embedding of an example $\mathbf{x}$. Then, the $(k, i)$-th component of the Fisher information embedding $\boldsymbol{f}(\boldsymbol{\theta}_t; \mathbf{x})$ can be formally expressed as*

$$\boldsymbol{f}(\boldsymbol{\theta}_t; \mathbf{x})_{k,i} = \sum_{y=1}^{K} p(y|\mathbf{x}; \boldsymbol{\theta}_t) \left[\nabla_{w_{ki}} \log p(y|\mathbf{x}; \boldsymbol{\theta}_t)\right]^2 = p_k(1 - p_k)\mathbf{h}(\boldsymbol{\theta}_t; \mathbf{x})_i^2, \tag{12}$$

*where $k \in [1, K], i \in [1, d]$, $p_k = \frac{\exp^{z_{\mathbf{x},k}}}{\sum_{j=1}^{K} \exp^{z_{\mathbf{x},j}}}$ the softmax probability of an example $\mathbf{x}$ for the class $k$, and $z_{\mathbf{x},k}$ the logit for the class $k$ of the example $\mathbf{x}$.*

*Proof.* The complete proof is available in Appendix C. □

### 4.4 FISHER-OPTIMALITY-PRESERVING PROPERTIES

Since the Fisher-based AL objective in Eq. (8) is not submodular (Ash et al., 2021), greedy optimization does not guarantee a bounded approximation. Furthermore, understanding Eq. (4) for multivariate models is challenging since it involves the inverse of the Fisher information matrix. Hence, it is difficult to get a clear intuition of what examples are most beneficial for optimizing the objective function. However, we show that by simplifying the objective function from a Fisher information matrix to a Fisher information embedding, we can get intuitions about what properties $\mathbf{x} \in S_t^*$ should possess. The objective function of Eq. (4) can be rewritten as

$$S_t^* = \underset{S_t \subset U_t, |S_t| \leq b}{\arg\min} \sum_{i=1}^{|\boldsymbol{\theta}_t|} \frac{t_i}{s_i}, \tag{13}$$

where the $i$-th components of the *target* Fisher information embedding $\boldsymbol{F}(\boldsymbol{\theta}_t; M_t, U_t)$ and the *candidate* Fisher information embedding $\boldsymbol{F}(\boldsymbol{\theta}_t; S_t)$ are $t_i$ and $s_i$, respectively. As the value of $t_i$ does not change, we find two properties that the $s_i$ should have under two different conditions.

**Property 1.** Position-Wise Optimality: If we fix all Fisher embeddings $s_i : \forall i \neq k$, a larger $s_k$ will be closer to optimizing Eq. (13). That is, having large information generally for all $s_i : \forall i \in [0, |\boldsymbol{\theta}_t|]$ is beneficial for optimization.

**Property 2.** Distribution-Wise Optimality: If we assume $\|\boldsymbol{F}(\boldsymbol{\theta}_t; S_t)\|_2 = k$ where $k$ is an arbitrary constant, we demonstrate through Theorem 4.3, that a subset $S_t$ whose candidate Fisher information embedding $\boldsymbol{F}(\boldsymbol{\theta}_t; S_t)$ has a similar distribution with that of the target Fisher information embedding $\boldsymbol{F}(\boldsymbol{\theta}_t; M_t, U_t)$ will be beneficial for Eq. (13).

**Theorem 4.3.** *Let $s_i$ be the $i$-th component of the Fisher information embedding of an arbitrary subset $S_t$. Under $\|\boldsymbol{F}(\boldsymbol{\theta}_t, S_t)\|_2 = \sqrt{\sum_{|\boldsymbol{\theta}_t|} |s_i|^2} = k$, the optimal $s_i$ that minimizes Eq. (13) is $s_i = ct_i^{1/3}$, where $c = \sqrt[3]{2}k/\sum_{i=1}^{|\boldsymbol{\theta}_t|} t_i^{2/3}$.*

*Proof.* The complete proof is available in Appendix D. □

### 4.5 PUTTING THEM ALL TOGETHER: ACCUACL

#### 4.5.1 GREEDY QUERY STRATEGY

We propose a novel greedy query strategy based on the two properties discussed in Section 4.4. AccuACL successfully constructs a batch of examples with the properties at task $t$.

**Magnitude Score.** Based on Property 1, in order to reward the examples with higher Fisher information, we propose the magnitude score $\mathcal{M}(\boldsymbol{\theta}_t, \mathbf{x})$, which is the $\ell_2$-norm of the Fisher information embedding $\mathcal{M}(\boldsymbol{\theta}_t, \mathbf{x}) = \|\boldsymbol{f}(\boldsymbol{\theta}_t; \mathbf{x})\|_2$.

**Distribution Score.** Based on Property 2, in order to reward the examples with similar information distribution to $\boldsymbol{F}(\boldsymbol{\theta}_t; M_t, U_t)$, we propose the distribution score $\mathcal{D}(\boldsymbol{\theta}_t, \mathbf{x}, M_t, U_t)$, which is the Jensen-Shannon divergence (Lin, 1991) between the distributions of $\boldsymbol{f}(\boldsymbol{\theta}_t; \mathbf{x})$ and $\boldsymbol{F}(\boldsymbol{\theta}_t; M_t, U_t)$,

$$\mathcal{D}(\boldsymbol{\theta}_t, \mathbf{x}, M_t, U_t) = \exp(-D_{\text{JS}}(\sigma(\boldsymbol{f}(\boldsymbol{\theta}_t; \mathbf{x}))\|\sigma(\boldsymbol{F}(\boldsymbol{\theta}_t; M_t, U_t)))), \quad (14)$$

where $\sigma(\mathbf{z}) = \frac{e^{\mathbf{z}}}{\sum_j e^{z_j}}$ is the softmax function, and $D_{\text{JS}}(\cdot\|\cdot)$ is the Jensen-Shannon divergence.

**Merger of the Two Scores.** In order to select the examples that satisfy both properties, we overly-sample the subset that ranks highest according to $\mathcal{D}(\cdot)$, and then further narrow it down by selecting its subset that ranks highest according to $\mathcal{M}(\cdot)$. Intuitively, after identifying important parameters for the past and the new through the target Fisher information $\boldsymbol{F}(\boldsymbol{\theta}_t; M_t, U_t)$, AccuACL prioritizes sample selection to preserve the stability of past parameters while effectively optimizing those important for the new task.

#### 4.5.2 COMPLEXITY ANALYSIS

**Space Complexity.** Owing to the Fisher information embedding, for each AL round, AccuACL has a space complexity of $O((m + n)dK)$, where $m$ is the memory buffer size, $n$ is the data pool size, $d$ is the embedding dimensionality, and $K$ is the total number of classes, which is significantly less than the space complexity of $O((m + n)d^2K^2)$ required for the query strategy of BAIT (Ash et al., 2021).

**Time Complexity.** For each AL round, AccuACL has a runtime complexity of $O(n \log n + dK \log dK)$, where $n \log n$ is induced by selecting the examples with the highest score, and $dK \log dK$ is induced by selecting the dimensions with the highest Fisher information for measuring $\mathcal{D}(\cdot)$. On the other hand, BAIT (Ash et al., 2021) has a time complexity of $O(bndK + n(dK)^2)$, where $b$ is the labeling budget, which is very expensive compared to AccuACL.

## 5 EXPERIMENT

### 5.1 EXPERIMENTAL SETUP

**Algorithms.** We compare AccuACL with six AL algorithms in combination with four rehearsal-based CL methods. The implementation details can be found in Appendix E.

- *Active learning*: (1) Uniform randomly selects examples from the unlabeled data at each task, (2) Entropy (Settles, 1995) selects the examples for which the model is the least certain—i.e., by selecting those with the highest entropy of predicted probability distribution, (3) LeastConfidence (Wang and Shang, 2014) selects the examples that have the smallest confidence on the highest probability

Table 1: Performance comparison of AL baselines and **AccuACL** combined with rehearsal-based CL methods on SplitCIFAR10, SplitCIFAR100, and SplitTinyImageNet. The best and the second-best results are in **bold** and underline, respectively.

| Continual Learning | Active Learning | SplitCIFAR10 | | | | SplitCIFAR100 | | | | SplitTinyImageNet | | | |
|---|---|---|---|---|---|---|---|---|---|---|---|---|---|
| | | M=100 | | M=200 | | M=500 | | M=1000 | | M=2000 | | M=5000 | |
| | | $A_5(\uparrow)$ | $F_5(\downarrow)$ | $A_5(\uparrow)$ | $F_5(\downarrow)$ | $A_{10}(\uparrow)$ | $F_{10}(\downarrow)$ | $A_{10}(\uparrow)$ | $F_{10}(\downarrow)$ | $A_{10}(\uparrow)$ | $F_{10}(\downarrow)$ | $A_{10}(\uparrow)$ | $F_{10}(\downarrow)$ |
| ER | Full | 20.1±0.6 | 93.3±1.2 | 26.3±3.5 | 85.9±4.4 | 12.6±0.1 | 75.0±0.7 | 17.9±0.3 | 68.5±0.6 | 7.7±0.1 | 60.4±0.6 | 11.5±0.2 | 54.1±0.5 |
| | Uniform | 20.4±3.0 | 77.1±4.8 | 26.7±3.0 | **67.2**±5.2 | 10.9±0.4 | 63.7±0.6 | 16.7±0.4 | 56.7±0.3 | 6.8±0.2 | 45.3±0.3 | 8.9±0.3 | 42.5±0.6 |
| | Entropy | 19.7±1.2 | 81.7±0.7 | 23.6±1.8 | 76.5±2.3 | 8.5±0.3 | 64.7±0.6 | 11.0±0.4 | 61.8±0.5 | 4.7±0.1 | 44.1±0.2 | 5.5±0.2 | 42.8±0.2 |
| | LeastConf | 19.9±1.4 | 81.0±0.6 | 22.5±1.7 | 76.4±0.8 | 8.8±0.1 | 66.3±0.2 | 11.3±0.3 | 63.5±0.1 | 4.8±0.3 | 44.4±0.7 | 5.7±0.1 | 42.7±0.7 |
| | kCenter | 19.4±1.2 | **76.2**±1.5 | 23.4±2.1 | 71.8±1.6 | 9.7±0.7 | 66.1±0.2 | 14.7±0.7 | 60.0±0.2 | 6.0±0.3 | 46.1±0.7 | 7.4±0.3 | 44.6±0.4 |
| | BADGE | 19.4±1.7 | 81.0±1.7 | 25.1±1.5 | 73.5±1.4 | 9.0±0.0 | 66.4±0.3 | 12.5±0.3 | 62.2±0.7 | 5.8±0.2 | 45.5±0.3 | 7.2±0.1 | 43.0±0.8 |
| | BAIT | 18.4 | 82.3 | 23.3 | 76.5 | * | * | * | * | * | * | * | * |
| | **AccuACL** | **20.7**±1.0 | 77.9±1.2 | **26.9**±2.0 | 70.3±0.1 | **14.1**±0.7 | 55.8±0.9 | **22.0**±1.1 | 44.5±1.5 | 7.3±0.0 | 41.9±0.3 | **10.5**±1.0 | 37.5±1.0 |
| GSS | Full | 22.9±0.3 | 88.9±0.6 | 27.8±2.6 | 82.0±3.4 | 10.1±0.6 | 67.9±0.5 | 10.8±0.7 | 67.3±1.2 | 7.2±0.3 | 54.5±0.4 | 8.0±0.4 | 53.0±1.2 |
| | Uniform | 19.7±1.0 | 76.7±2.8 | 23.6±1.9 | 71.7±2.1 | 7.9±0.4 | 57.6±1.6 | 7.9±0.3 | 57.4±0.3 | 5.3±0.1 | 42.0±0.2 | 5.3±0.2 | 42.1±0.2 |
| | Entropy | 18.0±0.6 | 75.4±3.1 | 17.1±1.5 | 76.4±3.0 | 7.0±0.3 | 57.2±0.6 | 7.3±0.3 | 56.1±0.2 | 4.0±0.2 | **39.1**±0.8 | 4.3±0.2 | 40.0±0.1 |
| | LeastConf | 18.4±1.4 | 77.8±3.3 | 20.6±1.6 | 72.0±5.5 | 7.1±0.1 | 58.2±0.7 | 7.2±0.2 | 57.1±0.3 | 3.9±0.2 | 40.0±1.3 | 4.3±0.3 | 39.8±1.7 |
| | kCenter | 19.1±0.6 | 77.8±1.7 | 19.6±0.8 | 75.1±3.3 | 7.1±0.5 | 59.3±1.2 | 7.5±0.6 | 56.2±4.6 | 5.1±0.2 | 41.2±1.2 | 5.1±0.3 | 42.1±0.4 |
| | BADGE | 18.6±0.9 | 78.6±1.9 | 20.6±1.7 | 74.1±5.8 | 7.7±0.5 | 57.7±0.6 | 7.4±0.7 | 57.5±2.0 | 4.5±0.3 | 40.4±0.7 | 4.5±0.2 | 41.3±0.7 |
| | BAIT | 17.5 | 81.8 | 16.6 | 76.8 | * | * | * | * | * | * | * | * |
| | **AccuACL** | **26.5**±0.7 | 68.2±2.2 | **30.0**±0.6 | 61.3±1.8 | **8.4**±0.4 | 53.7±1.7 | **8.4**±0.4 | 54.3±1.0 | 5.7±0.4 | 39.8±1.0 | 5.8±0.2 | 38.4±1.7 |
| DER++ | Full | 40.0±1.1 | 68.6±1.3 | 48.7±1.1 | 57.5±1.1 | 30.6±1.2 | 51.1±0.9 | 40.1±1.4 | 38.0±1.0 | 10.3±0.3 | 55.3±1.1 | 19.6±0.1 | 32.2±0.2 |
| | Uniform | 39.2±0.4 | 49.0±1.5 | 49.6±1.2 | 31.9±2.1 | 27.6±0.9 | 38.9±1.4 | 35.9±0.7 | 21.5±0.5 | 11.3±0.2 | 29.0±0.4 | 15.2±0.1 | 10.7±0.3 |
| | Entropy | 32.3±0.6 | 62.9±0.2 | 47.5±4.2 | 38.6±4.7 | 21.3±0.7 | 48.6±1.0 | 31.7±0.2 | 27.5±0.8 | 8.1±0.1 | 29.7±1.1 | 13.1±0.3 | 9.7±0.4 |
| | LeastConf | 33.8±4.2 | 62.1±5.6 | 45.2±2.6 | 42.1±3.3 | 22.1±0.6 | 48.0±1.5 | 33.1±1.0 | 27.0±0.6 | 8.5±0.3 | 28.9±0.7 | 13.3±0.6 | **9.5**±0.5 |
| | kCenter | 37.0±1.1 | 55.1±2.3 | 47.0±1.6 | 39.6±3.5 | 25.9±0.0 | 43.4±0.3 | 35.0±0.7 | 24.9±0.6 | 10.7±0.1 | 27.9±0.8 | 14.4±0.4 | 11.2±0.5 |
| | BADGE | 36.4±2.3 | 57.9±0.6 | **51.0**±2.8 | 35.8±3.1 | 24.8±0.4 | 45.6±1.0 | 34.1±1.0 | 27.7±0.7 | 9.7±0.1 | 28.5±0.8 | 14.7±0.2 | 10.8±0.2 |
| | BAIT | 36.7 | 56.5 | 49.7 | 36.4 | * | * | * | * | * | * | * | * |
| | **AccuACL** | **44.2**±4.6 | 40.4±6.1 | 50.1±2.6 | 28.1±1.8 | **30.5**±0.2 | 27.0±0.4 | **36.3**±0.4 | 15.0±0.5 | 12.5±0.4 | 24.0±0.8 | **15.7**±0.6 | 11.4±0.3 |
| ACE | Full | 57.6±1.2 | 27.9±0.3 | 63.7±0.5 | 22.4±1.0 | 34.9±1.2 | 34.6±0.8 | 40.1±0.7 | 30.5±0.8 | 16.8±0.4 | 36.5±0.7 | 20.2±0.3 | 30.9±0.2 |
| | Uniform | 41.3±1.3 | 25.9±1.8 | **49.6**±1.6 | 20.0±3.6 | **28.4**±0.4 | 30.0±0.6 | **34.2**±0.6 | 25.5±1.1 | 12.3±1.0 | 27.6±0.9 | 14.6±0.2 | 23.2±0.5 |
| | Entropy | 42.7±1.0 | 30.3±1.6 | 47.0±2.5 | 28.8±2.9 | 24.9±0.4 | 37.1±0.6 | 31.5±0.5 | 31.4±0.7 | 9.5±0.4 | 27.8±0.5 | 11.9±0.2 | 23.4±0.6 |
| | LeastConf | 41.8±2.4 | 32.5±1.1 | 47.4±1.6 | 27.8±2.4 | 25.7±0.4 | 35.7±0.3 | 30.9±1.0 | 31.7±0.6 | 9.9±0.3 | 28.5±0.2 | 11.8±0.2 | 23.9±0.6 |
| | kCenter | 36.8±0.6 | 33.0±2.2 | 43.2±3.0 | 29.6±4.6 | 27.4±0.6 | 34.0±0.7 | 33.1±0.7 | 28.8±0.9 | 11.4±0.2 | 29.4±0.5 | 13.7±0.3 | 25.3±0.1 |
| | BADGE | 41.3±2.4 | 32.3±1.9 | 47.8±0.9 | 26.7±0.7 | 26.5±0.3 | 35.9±0.4 | 33.4±0.7 | 30.3±0.7 | 11.1±0.4 | 28.5±0.5 | 13.5±0.3 | 24.5±0.5 |
| | BAIT | 41.2 | 33.3 | 48.0 | 28.3 | * | * | * | * | * | * | * | * |
| | **AccuACL** | **43.7**±1.7 | 20.8±1.3 | 48.1±1.1 | 17.7±1.4 | **28.4**±0.6 | 26.1±0.4 | 33.9±1.0 | 20.9±1.2 | **13.4**±0.1 | 24.9±0.5 | **16.1**±0.4 | 20.5±0.2 |

* Out of memory during execution.

class, (4) kCenterGreedy (Sener and Savarese, 2017) tries to select the most diverse examples that are farthest from each other in feature space, (5) BADGE (Ash et al., 2019) is a hybrid approach that selects both diverse and uncertain examples, (6) BAIT (Ash et al., 2021) reduces the complexity of Fisher-based AL, making it suitable for use in deep learning environments, and Full is the fully-supervised setting.

- *Continual learning*: (1) ER (Rolnick et al., 2019) stores random examples from previous tasks, (2) GSS (Aljundi et al., 2019) stores the examples that can diversify the gradients, (3) DER++ (Buzzega et al., 2020) further uses knowledge distillation to enhance stability, and (4) ER-ACE (Caccia et al., 2022) reduces abrupt changes on representations to discourage disruptive parameter updates.

**Datasets.** We use three popular CL benchmark datasets in class-incremental scenarios, SplitCIFAR10, SplitCIFAR100, and SplitTinyImageNet, which are all derived from the original CIFAR10, CIFAR100 (Alex, 2009), and TinyImageNet (Le and Yang, 2015), respectfully. SplitCIFAR10 splits 50K training images in CIFAR10 into five tasks, where each task includes 10K images for 2-way classification. SplitCIFAR100 splits 50K training images in CIFAR100 into ten tasks, where each task includes 5K images of 10-way classification. SplitTinyImageNet divides 100K training images in TinyImageNet into ten tasks, where each task involves 10K images of 20-way classification.

**Metrics.** We use two commonly-used metrics for CL, average accuracy ($A_T$) and forgetting ($F_T$) (De Lange et al., 2021). The *average accuracy* $A_T = \frac{1}{T}\sum_{j=1}^{T} a_{T,j}$ averages all the test accuracy $a_{T,j}$, which is the accuracy of the $j$-th task measured after learning all $T$ tasks. On the other hand, the *forgetting* $F_T = \frac{1}{T-1}\sum_{j=1}^{T-1} f_{j,T}$ averages all forgetting $f_{j,T}$, where $f_{j,T}$ measures the difference between the accuracy of task $j$ measured after learning task $j$ and task $T$.

## 5.2 MAIN RESULTS

**Overall Performance.** Table 1 shows the overall performance of AccuACL and six AL baselines, combined with four rehearsal-based CL methods. Overall, AccuACL achieves the best performance

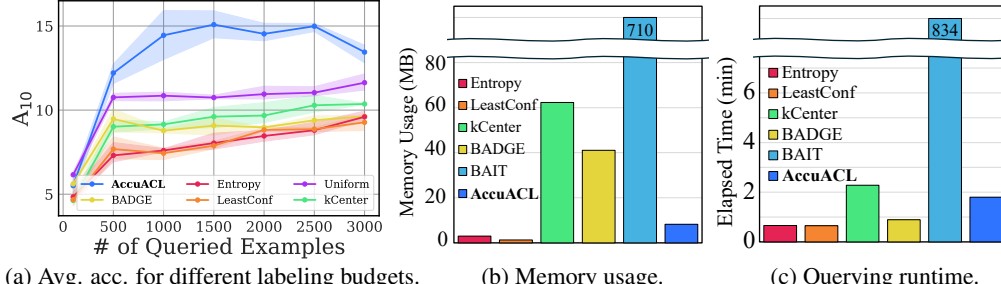

Figure 2: Comparison of AL strategies: (a) average accuracy of SplitCIFAR100 on ER for different labeling budget per task; (b) memory consumed for selecting 100 examples for a single task in SplitCIFAR10; (c) elapsed querying time for selecting 1000 examples for every task in SplitCIFAR10.

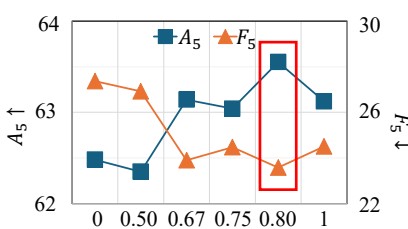

Figure 3: Effect of the parameter $\lambda$ for SplitCIFAR10 on task 4.

Table 2: Performance of baseline AL methods with memory buffer examples as an initially selected subset.

| AL Method | SplitCIFAR100 | | SplitTinyImageNet | |
|---|---|---|---|---|
| | $A_{10}(\uparrow)$ | $F_{10}(\downarrow)$ | $A_{10}(\uparrow)$ | $F_{10}(\downarrow)$ |
| kCenter | $27.4_{\pm0.6}$ | $34.0_{\pm0.7}$ | $11.4_{\pm0.2}$ | $29.4_{\pm0.5}$ |
| kCenter + Mem | $26.5_{\pm0.6}$ | $35.3_{\pm0.7}$ | $11.1_{\pm0.3}$ | $29.7_{\pm0.9}$ |
| BADGE | $26.5_{\pm0.3}$ | $35.9_{\pm0.4}$ | $11.1_{\pm0.4}$ | $28.5_{\pm0.5}$ |
| BADGE + Mem | $26.2_{\pm0.7}$ | $36.0_{\pm0.3}$ | $11.0_{\pm0.3}$ | $29.0_{\pm0.3}$ |
| **AccuACL** | $\mathbf{28.4}_{\pm0.6}$ | $\mathbf{26.1}_{\pm0.4}$ | $\mathbf{13.4}_{\pm0.1}$ | $\mathbf{24.9}_{\pm0.5}$ |

in most cases, outperforming other AL baselines by $23.8\%$ and $17.0\%$ in average accuracy and forgetting, respectively in average. This superior performance indicates that considering the ability to prevent forgetting is as significant as the ability to acquire new knowledge quickly in query strategy to boost the performance in ACL. On the other hand, most AL baselines that focus only on learning new tasks actually under-perform even Uniform because they are biased too much towards new tasks, confirming our motivation. Also, BAIT is not scalable to large datasets with many classes, including SplitCIFAR100 and SplitTinyImageNet, because of its expensive query strategy. In contrast, owing to our efficient query algorithm, AccuACL succeeds in scaling to these datasets. For further analysis on the impact of task ordering, please refer to Appendix F.1.

Moreover, Figure 2(a) illustrates the performance trend of average accuracy as the per-task labeling budget varies. The results consistently show that AccuACL outperforms the AL baselines across different labeling budget settings, highlighting its robustness and adaptability. Interestingly, AccuACL reaches the optimal point even without using the entire dataset for training, which is also shown in several settings in Table 1. This phenomenon aligns with findings in core-set selection for CL (Yoon et al., 2022), where the goal is to select high-quality subsets from a larger pool of data to enhance performance. In the context of ACL, AccuACL can be viewed as selecting high-quality examples in an *unsupervised manner*. The experimental settings in which ACL surpasses CL are mostly attributed to AccuACL, indicating AccuACL's effectiveness in identifying example-wise quality. For further analysis on the labeling budget in a single task, please refer to Appendix F.2.

**Efficiency.** Figures 2(b) and 2(c) depict the memory usage and time spent for querying, respectively, for different AL algorithms, tested on SplitCIFAR10($M$=100). Due to our effective approximation in Section 4.3 and querying algorithm in Section 4.5, AccuACL can perform querying with reasonable time and space consumption. Conversely, BAIT requires 462.1 times more time and 85.5 times more space resources than AccuACL.

## 5.3 MORE RESULTS AND ABLATION STUDIES

**Effect of $\lambda$.** Figure 3 illustrates the influence of $\lambda$ on the fourth task of SplitCIFAR10, trained with ER, which is the parameter that dictates the quality of data to prioritize during query selection. The different values of $\lambda$ are selected by our theoretical value in Theorem 4.1 for different tasks in SplitCIFAR10. For a fair comparison, we choose identical initial points for the first round of training, select subsets for various $\lambda$ values, and then re-train it to examine the impact of a change in $\lambda$. It is

Table 3: Learning accuracy($LA_{10}$), forgetting($F_{10}$), and average accuracy($A_{10}$) comparison between AL baselines and AccuACL on SplitCIFAR100 trained with DER++.

| AL Method | Uniform | Entropy | LeastConf | kCenter | BADGE | **AccuACL** |
|---|---|---|---|---|---|---|
| $LA_{10}(\uparrow)$ | $62.2_{\pm0.6}$ | $64.9_{\pm0.2}$ | $65.8_{\pm1.3}$ | $65.0_{\pm0.3}$ | $\mathbf{65.9}_{\pm0.5}$ | $56.6_{\pm0.3}$ |
| $F_{10}(\downarrow)$ | $38.7_{\pm1.3}$ | $46.9_{\pm0.6}$ | $48.9_{\pm1.1}$ | $43.4_{\pm0.3}$ | $45.6_{\pm1.0}$ | $\mathbf{29.3}_{\pm0.1}$ |
| $A_{10}(\uparrow)$ | $27.4_{\pm0.6}$ | $22.6_{\pm0.4}$ | $21.7_{\pm0.3}$ | $25.9_{\pm0.0}$ | $24.8_{\pm0.4}$ | $\mathbf{30.0}_{\pm0.4}$ |

observed that the forgetting rises when $\lambda$ is small because AccuACL focuses on the new task, and vice versa. Furthermore, the accuracy reaches the optimal at the theoretical value of $\lambda$.

**AL Baselines with Memory.** To show how effectively AccuACL uses the information of past data via a memory buffer, we run experiments on the greedy algorithms, kCenter and BADGE, to use the examples in the memory buffer as an initially selected subset to incorporate the past knowledge into the algorithms. Surprisingly, as in Table 2, using the memory buffer as a starting point even degrades the performance of the original AL algorithms. We conjecture that additionally using the memory buffer may force AL algorithms to concentrate on learning new information even more as they attempt to choose the examples that are the most dissimilar from past data. This finding indicates that simply providing AL with past data is not sufficient, whereas AccuACL properly balances past and new knowledge through data-driven information. For further analysis on the impact of using a memory buffer in AccuACL, please refer to Appendix F.3.

**Learning Accuracy in ACL.** *Learning accuracy* (Mirzadeh et al., 2022) is an essential evaluation for assessing the plasticity of CL algorithms, which is the accuracy of the new task immediately after learning. Within the context of ACL, it indicates the efficacy of AL algorithms in quickly adapting to new tasks. Accordingly, we perform an additional experiment evaluating the learning accuracy of AL strategies on the SplitCIFAR100 dataset($M$=500), optimized by DER++. As can be seen in Table 3, traditional AL baselines achieve superior learning accuracy, consistent with our motivation that traditional AL approaches emphasize *fast adaptation to new tasks*. On the other hand, AccuACL surpasses the AL baselines in forgetting, as AccuACL specifically tackles another essential perspective of ACL learning properties: *prevention of catastrophic forgetting*. By successfully balancing these two learning properties, AccuACL exhibits state-of-the-art performance in average accuracy, highlighting its value in overall ACL performance. Moreover, as AccuACL selects examples that mitigate recency bias in linear classifiers (Mai et al., 2021), it allows reliable Fisher information matrix calculation for the upcoming AL rounds.

**Scoring Methods.** Table 4 shows the effect of different scoring methods defined in Section 4.5. Querying based solely on $\mathcal{M}(\cdot)$ focuses exclusively on learning new information, neglecting the target Fisher information defined in Section 4.2. This approach leads to high forgetting and low accuracy. In contrast, querying based solely on $\mathcal{D}(\cdot)$ emphasizes the distribution of information across parameters. While this approach selects examples that balance information between past and new tasks, it neglects the amount of each example's informativeness. As a result, although it may reduce forgetting, it does not greatly enhance average accuracy. AccuACL employs $\mathcal{D}(\cdot)$ as the primary selection criterion for querying, aligning with the objective of our study to sustain a balance between the two learning properties of ACL—preventing catastrophic forgetting and quickly learning new tasks. Ensuring the magnitude of informativeness after establishing the balance has shown effectiveness through empirical evaluation.

Table 4: Effect of scoring measures.

| Score | SplitCIFAR100 | |
|---|---|---|
| | $A_{10}(\uparrow)$ | $F_{10}(\downarrow)$ |
| $\mathcal{M}(\cdot)$ | $33.0_{\pm1.4}$ | $28.4_{\pm0.9}$ |
| $\mathcal{D}(\cdot)$ | $\underline{33.2}_{\pm0.8}$ | $\mathbf{20.2}_{\pm0.3}$ |
| **AccuACL** | $\mathbf{33.9}_{\pm0.9}$ | $\underline{20.9}_{\pm1.0}$ |

## 6 CONCLUSION

In this paper, we introduce a new perspective for the query strategy in ACL, by effectively balancing the prevention of forgetting and quick learning of new tasks, even with limited access to the data for past tasks. Consequently, we propose **AccuACL**, leveraging the Fisher information matrix to efficiently convey the learned knowledge across tasks and precisely measure the new knowledge without labels. Extensive experiments confirm that AccuACL substantially improves many CL methods in ACL scenarios.

## ACKNOWLEDGMENTS

This work was supported by Institute of Information & Communications Technology Planning & Evaluation (IITP) grant funded by the Korea government (MSIT) (No. RS-2020-II200862, DB4DL: High-Usability and Performance In-Memory Distributed DBMS for Deep Learning, 50% and No. RS-2022-II220157, Robust, Fair, Extensible Data-Centric Continual Learning, 50%). Jae-Gil Lee was partly supported by Samsung Electronics Co., Ltd. (IO201211-08051-01).

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

# Active Learning for Continual Learning:
# Keeping the Past Alive in the Present

# (Appendix)

## A    COMPLETE PROOF OF THEOREM 4.1

**Theorem 4.1. Restated.** Let $U_{1:t}$ be the unlabeled data pool for all seen tasks until task $t$. Then, the *target* Fisher information matrix can be divided into past and new information matrices as

$$I(\boldsymbol{\theta}_t; U_{1:t}) = \lambda \cdot \underbrace{I(\boldsymbol{\theta}_t; U_{1:t-1})}_{\text{past information}} + (1 - \lambda) \cdot \underbrace{I(\boldsymbol{\theta}_t; U_t)}_{\text{new information}} \tag{15}$$

$$\tag{16}$$

with optimal value of the balancing parameter $\lambda = \frac{|U_{1:t-1}|}{|U_{1:t}|}$.

*Proof.*

$$I(\boldsymbol{\theta}_t; U_{1:t}) = -\frac{1}{|U_{1:t}|} \sum_{x \in U_{1:t}} \sum_{y=1}^{K} p(y|x; \boldsymbol{\theta}_t) \nabla_{\boldsymbol{\theta}_t}^2 p(y|x; \boldsymbol{\theta}_t) \tag{17}$$

$$= -\frac{|U_{1:t-1}|}{|U_{1:t}|} \frac{1}{|U_{1:t-1}|} \sum_{x \in U_{1:t-1}} \sum_{y=1}^{K} p(y|x; \boldsymbol{\theta}_t) \nabla_{\boldsymbol{\theta}_t}^2 p(y|x; \boldsymbol{\theta}_t) \tag{18}$$

$$- \frac{|U_t|}{|U_{1:t}|} \frac{1}{|U_t|} \sum_{x \in U_t} \sum_{y=1}^{K} p(y|x; \boldsymbol{\theta}_t) \nabla_{\boldsymbol{\theta}_t}^2 p(y|x; \boldsymbol{\theta}_t) \tag{19}$$

$$= \frac{|U_{1:t-1}|}{|U_{1:t}|} I(\boldsymbol{\theta}_t; U_{1:t-1}) + \frac{|U_t|}{|U_{1:t}|} I(\boldsymbol{\theta}_t; U_t), \quad \lambda = \frac{|U_{1:t-1}|}{|U_{1:t}|} \tag{20}$$

$\square$

## B    ANALYSIS OF THE APPROXIMATION IN SECTION 4.2

In Theorem B.1 , we assess the effectiveness of our approximation $I(\boldsymbol{\theta}_t; M_t, U_t)$ in comparison to the true $I(\boldsymbol{\theta}_t; U_{1:t})$. This theorem indicates that the difference between their variances decreases as the size of a memory buffer increases.

**Theorem B.1.** *Let $U_t$ be the unlabeled data pool for a task $t \in \{1, \ldots, |\mathcal{T}|\}$ and $M_t$ be the memory buffer $M_t \subset U_{1:t-1}$ of a certain rehearsal-based CL algorithm. Then, the difference of variance between two empirical Fisher information matrices $I(\boldsymbol{\theta}_t; U_{1:t})$ and $I$ is $\frac{|U_{1:t-1}|}{|U_{1:t}|^2} \left( \frac{|U_{1:t-1}| - |M_t|}{|M_t|} \right) \sigma_1^2$, where $\sigma_1^2$ is the inherent variance of the Fisher information of past tasks $\{1, \ldots, t-1\}$.*

*Proof.* Based on the optimal value of $\lambda$ derived from Theorem 4.1, we have

$$I(\boldsymbol{\theta}_t; U_{1:t}) = \frac{|U_{1:t-1}|}{|U_t|} \cdot I(\boldsymbol{\theta}_t; U_{1:t-1}) + \left( 1 - \frac{|U_{1:t-1}|}{|U_t|} \right) \cdot I(\boldsymbol{\theta}_t; U_t) \tag{21}$$

$$I(\boldsymbol{\theta}_t; M_t, U_t) = \frac{|U_{1:t-1}|}{|U_t|} \cdot I(\boldsymbol{\theta}_t; M_t) + \left( 1 - \frac{|U_{1:t-1}|}{|U_t|} \right) \cdot I(\boldsymbol{\theta}_t; U_t). \tag{22}$$

As we are relying on the Monte Carlo assumption of the Fisher information matrices (Sourati et al., 2017), we can employ the central limit theorem to assess the reliability of its estimates in relation to the ground truth. Given that two datasets from past and new tasks are independent, we have

$$\text{Var}[I(\boldsymbol{\theta}_t; U_{1:t})] = \left( \frac{|U_{1:t-1}|}{|U_t|} \right)^2 \cdot \frac{\sigma_1^2}{|U_{1:t-1}|} + \left( 1 - \frac{|U_{1:t-1}|}{|U_t|} \right)^2 \cdot \frac{\sigma_2^2}{|U_t|} \tag{23}$$

$$\text{Var}[I(\boldsymbol{\theta}_t; M_t, U_t)] = \left( \frac{|U_{1:t-1}|}{|U_t|} \right)^2 \cdot \frac{\sigma_1^2}{|M_t|} + \left( 1 - \frac{|U_{1:t-1}|}{|U_t|} \right)^2 \cdot \frac{\sigma_2^2}{|U_t|}, \tag{24}$$

where $\sigma_1^2$ and $\sigma_2^2$ are its inherent variance of Fisher information of the past tasks and the new task, respectively. Then, we can show that the discrepancy of two estimations in the perspective of its variance can be formulated as

$$\text{Var}\left[\boldsymbol{I}\left(\boldsymbol{\theta}_t; U_{1:t}\right)\right] - \text{Var}\left[\boldsymbol{I}\left(\boldsymbol{\theta}_t; M_t, U_t\right)\right] = \frac{|U_{1:t-1}|}{|U_{1:t}|^2}\left(\frac{|U_{1:t-1}| - |M_t|}{|M_t|}\right)\sigma_1^2, \tag{25}$$

which converges to 0 when the cardinality of $M_t$ reaches that of $U_{1:t-1}$. $\qquad\square$

## C COMPLETE PROOF OF THEOREM 4.2

**Lemma C.1.** *For an arbitrary example $\mathbf{x}$, let $K$ be the number of classes, $\boldsymbol{\theta}_t^{[L]} = (w_{11}, \ldots, w_{Kd}) \in \mathbb{R}^{Kd}$ be the parameters of the last linear classification layer, and $\mathbf{h}(\boldsymbol{\theta}_t; \mathbf{x}) \in \mathbb{R}^d$ be the embedding of an example $\mathbf{x}$. The logit $z \in \mathbb{R}^K$ of $\mathbf{x}$ is $z_c = \boldsymbol{\theta}_c^{[L]T} \cdot \mathbf{h}(\boldsymbol{\theta}_t; \mathbf{x})$ and the log-likelihood of $\mathbf{x}$ belonging to a class $c$ is $\ell_c = \log p(c|\mathbf{x}, \boldsymbol{\theta}_t) = \log p_c$, where $p_c = \sigma(z)_c = \frac{\exp^{z_c}}{\sum_{j=1}^K \exp^{z_j}}$ is the probability of $\mathbf{x}$ belonging to the class $c \in [1, K]$. Then, for the example $\mathbf{x}$, the $(i, j)$-th component of the gradient with respect to $\boldsymbol{\theta}_t^{[L]}$ is*

$$\frac{\partial \ell_c}{\partial w_{ij}} = (\mathbb{1}[i = c] - p_i)\,\mathbf{h}(\boldsymbol{\theta}; \mathbf{x})_j. \tag{26}$$

*Proof.* A similar derivation can be found in Ash et al. (2019).

$$\frac{\partial \ell_c}{\partial w_{ij}} = \frac{\partial \log p_c}{\partial z_i} \cdot \frac{\partial z_i}{\partial w_{ij}} \tag{27}$$

$$= \frac{\partial \log p_c}{\partial p_c} \cdot \frac{\partial p_c}{\partial z_i} \cdot \frac{\partial z_i}{\partial w_{ij}} \tag{28}$$

$$= \frac{1}{p_c} \cdot p_c(\mathbb{1}[i = c] - p_i) \cdot \mathbf{h}(\boldsymbol{\theta}; \mathbf{x})_j \tag{29}$$

$$= (\mathbb{1}[i = c] - p_i)\,\mathbf{h}(\boldsymbol{\theta}; \mathbf{x})_j. \tag{30}$$

$\square$

**Theorem 4.2. Restated.** *Let $K$ be the number of classes, $d$ be the number of embedding dimensions, $\boldsymbol{\theta}_t^{[L]} = (w_{11}, \ldots, w_{Kd}) \in \mathbb{R}^{Kd}$ be the parameters of the last linear classification layer, and $\mathbf{h}(\boldsymbol{\theta}_t; \mathbf{x}) \in \mathbb{R}^d$ be the embedding of an example $\mathbf{x}$. Then, the $(k, i)$-th component of the Fisher information embedding $\boldsymbol{f}(\boldsymbol{\theta}_t; \mathbf{x})$ can be formally expressed as*

$$\boldsymbol{f}(\boldsymbol{\theta}_t; \mathbf{x})_{k,i} = \sum_{y=1}^K p(y|\mathbf{x}; \boldsymbol{\theta}_t)\left[\nabla_{w_{ki}} \log p(y|\mathbf{x}; \boldsymbol{\theta}_t)\right]^2 = p_k(1 - p_k)\mathbf{h}(\boldsymbol{\theta}_t; \mathbf{x})_i^2, \tag{31}$$

*where $k \in [1, K], i \in [1, d]$, $p_k = \frac{\exp^{z_{\mathbf{x},k}}}{\sum_{j=1}^K \exp^{z_{\mathbf{x},j}}}$ the softmax probability of an example $\mathbf{x}$ for the class $k$, and $z_{\mathbf{x},k}$ the logit for the class $k$ of the example $\mathbf{x}$.*

*Proof.* The Fisher information matrix is defined as the covariance of a score function. When we consider only diagonal components, it can be formally expressed as Eq. (32), since the expectation of the score function is 0 (Sourati et al., 2017). Using the results in Lemma C.1, we can derive the

Fisher embedding as

$$f(\boldsymbol{\theta}_t; \mathbf{x})_{k,i} = \sum_{y=1}^{K} p(y|\mathbf{x}, \boldsymbol{\theta}_t) \left[ \nabla_{w_{ki}} \log p(y|\mathbf{x}, \boldsymbol{\theta}_t) \right]^2 \tag{32}$$

$$= \sum_{y=1}^{K} p_y \left( \frac{\partial L_y}{\partial w_{ki}} \right)^2 \tag{33}$$

$$= \sum_{y=1}^{K} \left[ p_y \left( \mathbb{1}[k=y] - p_k \right)^2 \right] \cdot \mathbf{h}(\boldsymbol{\theta}; \mathbf{x})_i^2 \tag{34}$$

$$= \left[ p_k(1-p_k)^2 + p_k^2 \left( \sum_{y \neq k} p_y \right) \right] \cdot \mathbf{h}(\boldsymbol{\theta}; \mathbf{x})_i^2 \tag{35}$$

$$= \left[ p_k(1-p_k)^2 + p_k^2(1-p_k) \right] \cdot \mathbf{h}(\boldsymbol{\theta}; \mathbf{x})_i^2 \tag{36}$$

$$= p_k(1-p_k)\mathbf{h}(\boldsymbol{\theta}; \mathbf{x})_i^2. \tag{37}$$

$\square$

## D  COMPLETE PROOF OF THEOREM 4.3

**Theorem 4.3. Restated.** *Let $s_i$ be the $i$-th component of the Fisher information embedding of an arbitrary subset $S_t$. Under $\|\boldsymbol{F}(\boldsymbol{\theta}_t, S_t)\|_2 = \sqrt{\sum_{|\boldsymbol{\theta}_t|} |s_i|^2} = k$, the optimal $s_i$ that minimizes Eq. (13) is $s_i = ct_i^{1/3}$, where $c = \sqrt[3]{2}k/\sum_{i=1}^{|\boldsymbol{\theta}_t|} t_i^{2/3}$.*

*Proof.* We aim to find a subset $S_t$ that satisfies $\arg\min_{S_t} \sum_{i=1}^{|\boldsymbol{\theta}_t|} \frac{t_i}{s_i}$, when $\|\boldsymbol{F}(\boldsymbol{\theta}_t, S_t)\|_2 = k, k \in \mathbb{R}^+$. As we aim to find the minimum of $f(s_1, \ldots, s_{|\boldsymbol{\theta}_t|}) = \sum_{i=1}^{|\boldsymbol{\theta}_t|} t_i/s_i$ with a constraint $\|\boldsymbol{F}(\boldsymbol{\theta}_t, S_t)\|_2 = k$, Lagrange multipliers can be employed. To solve the Lagrange multiplier method, we define $f(s_1, \ldots, s_{|\boldsymbol{\theta}_t|})$, the function that we wish to minimize, and $g(s_1, \ldots, s_{|\boldsymbol{\theta}_t|})$, the constraint, by

$$g(s_1, \ldots, s_{|\boldsymbol{\theta}_t|}) = \sum_{i=1}^{|\boldsymbol{\theta}_t|} s_i^2 - k^2 = 0, \tag{38}$$

$$f(s_1, \ldots, s_{|\boldsymbol{\theta}_t|}) = \sum_{i=1}^{|\boldsymbol{\theta}_t|} \frac{t_i}{s_i}. \tag{39}$$

Then, we are able to construct the Lagrangian function $L(s_1, \ldots, s_{|\boldsymbol{\theta}_t|}, \lambda)$ as

$$L(s_1, \ldots, s_{|\boldsymbol{\theta}_t|}, \lambda) = f(s_1, \ldots, s_{|\boldsymbol{\theta}_t|}) - \lambda g(s_1, \ldots, s_{|\boldsymbol{\theta}_t|}). \tag{40}$$

Then, we find the extrema for every element $s_i$, $\forall i \in \{1, \ldots, |\boldsymbol{\theta}_t|\}$ by taking the partial derivative of $L$ as

$$L(s_1, \ldots, s_{|\boldsymbol{\theta}_t|}, \lambda) = \sum_{i=1}^{|\boldsymbol{\theta}_t|} \frac{t_i}{s_i} - \lambda \left( \sum_{i=1}^{|\boldsymbol{\theta}_t|} s_i^2 - k^2 \right) \tag{41}$$

$$\frac{\partial f(s_1, \ldots, s_{|\boldsymbol{\theta}_t|})}{\partial s_i} = -\frac{s_i^2}{t_i} - 2s_i\lambda = 0, \ \forall i \in \{1, \ldots, |\boldsymbol{\theta}_t|\} \tag{42}$$

$$s_i = (-\frac{t_i}{2\lambda})^{1/3}. \tag{43}$$

Finally, we use each value of $s_i$ to find the value of the Lagrange multiplier $\lambda$ to substitute and finalize the proof.

$$\sum_{i=1}^{|\boldsymbol{\theta}_t|} (\frac{t_i}{2\lambda})^{2/3} = k \tag{44}$$

$$\sum_{i=1}^{|\boldsymbol{\theta}_t|} \frac{t_i^{2/3}}{k} = (2\lambda)^{2/3} \tag{45}$$

$$\lambda = -\frac{(\sum_{i=1}^{|\boldsymbol{\theta}_t|} t_i^{2/3})^{3/2}}{2k^{3/2}} (\because s_i \geq 0) \tag{46}$$

$$s_i = \frac{\sqrt{k} t_i^{1/3}}{\sum_{i=1}^{|\boldsymbol{\theta}_t|} t_i^{2/3}} \tag{47}$$

$$\therefore s_i = C t_i^{1/3} \tag{48}$$

$\square$

## E    IMPLEMENTATION DETAILS

Table 5: Detailed experiment settings.

| Hyperparamters | | SplitCIFAR10 | SplitCIFAR100 | SplitTinyImageNet |
|---|---|---|---|---|
| | architecture | ResNet18 | ResNet18 | ResNet18 |
| | training epoch | 50 | 20 | 30 |
| | batch size | 16 | 16 | 32 |
| Training | optimizer | SGD | SGD | SGD |
| Configuration | momentum | 0.8 | 0.8 | 0.9 |
| | weight decay | $10^{-4}$ | $10^{-4}$ | 0 |
| | learning rate (lr) | 0.01 | 0.01 | 0.01 |
| | lr scheduler | Cosine Annealing | Cosine Annealing | Cosine Annealing |

For each CL task, we perform multi-round ACL following the conventional AL setup (Ash et al., 2019). For each experiment, we use two different sizes for a memory buffer, $\{100, 200\}$ for SplitCIFAR10, $\{500, 1000\}$ for SplitCIFAR100, and $\{2000, 5000\}$ for SplitTinyImageNet. We query 1000, 2000, and 3000 examples during ten rounds of AL for SplitCIFAR10, SplitCIFAR100, and SplitTinyImageNet for each task, respectively. In terms of over-sampling in Section 4.5, we over-sample two times the original query size.

The overall optimization and training setups are shown in Table 5. Among the four CL algorithms, GSS (Aljundi et al., 2019) and DER (Buzzega et al., 2020) require hyperparameters to fix. For GSS, the number of random samples for determining the maximal cosine similarity is set to 5. For DER, the values of $\alpha$ and $\beta$ are given weights of 0.1 and 0.5, respectively. $\alpha$ represents the weight allocated to the mean-squared error, while $\beta$ represents the weight assigned to the cross-entropy error. All AL baselines, except for BAIT (Ash et al., 2021), need no additional hyperparameters to be specified. The oversampling rate for forward greedy optimization is configured as 2. For AccuACL, we choose the top-10 embeddings per class for SplitCIFAR10 and the top-5 embeddings for SplitCIFAR100 and SplitTinyImageNet in order to compute the distribution score. We conduct our experiment based on the Avalanche codebase (Carta et al., 2023). All experiments are implemented with PyTorch 1.12.1 and performed with NVIDIA GeForce RTX 4090 24GB on CUDA version 12.0. All experiments except for BAIT are repeated three times, and the average and standard deviation are reported.

## F    EXTENDED EXPERIMENTS

### F.1    ROBUSTNESS OF ACCUACL WITH VARYING TASK ORDERS

To further verify the robustness of our technique, we run additional experiments on the SplitCIFAR100($M$=500) dataset, where the original data is split into ten tasks of ten classes each

Table 6: Performance comparison of AccuACL and Uniform across multiple task order permutations on SplitCIFAR100($M$=500), trained with ER.

| AL Method | Sequential | | Perm. #1 | | Perm. #2 | | Perm. #3 | | Perm. #4 | |
|---|---|---|---|---|---|---|---|---|---|---|
| | $A_{10}(\uparrow)$ | $F_{10}(\downarrow)$ | $A_{10}(\uparrow)$ | $F_{10}(\downarrow)$ | $A_{10}(\uparrow)$ | $F_{10}(\downarrow)$ | $A_{10}(\uparrow)$ | $F_{10}(\downarrow)$ | $A_{10}(\uparrow)$ | $F_{10}(\downarrow)$ |
| Uniform | 10.9±0.4 | 63.7±0.6 | 11.7±0.4 | 62.0±0.3 | 11.1±0.2 | 61.0±1.2 | 11.9±0.4 | 62.5±0.5 | 11.9±0.4 | 61.4±0.5 |
| **AccuACL** | **14.1**±0.7 | **55.8**±0.9 | **14.7**±1.1 | **52.8**±1.1 | **14.6**±0.5 | **52.4**±0.6 | **15.4**±0.8 | **53.9**±1.0 | **15.8**±0.4 | **51.2**±0.4 |

in sequential order. To investigate the impact of task order, we generate four random permutations of the tasks. Table 6 demonstrates that AccuACL consistently outperforms the second-best method, Uniform, in all permutations. Our results indicate that our method is highly robust to changes in task order, further establishing its efficacy.

## F.2 PERFORMANCE ANALYSIS OF AL STRATEGIES THROUGHOUT AL ROUNDS

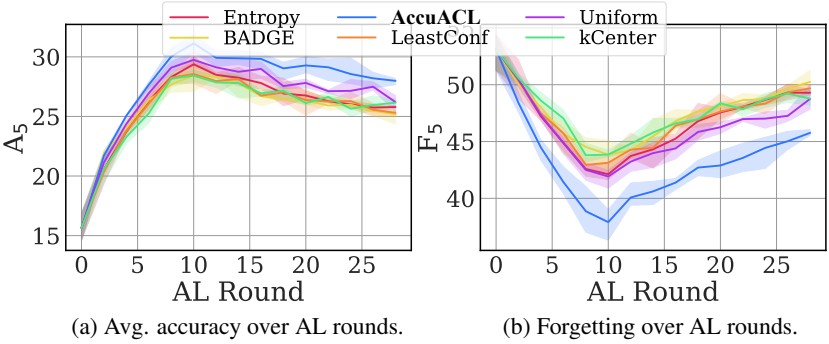

(a) Avg. accuracy over AL rounds.  (b) Forgetting over AL rounds.

Figure 4: Comparison of AL strategies: (a) average accuracy of SplitCIFAR100 on ER throughout AL rounds; (b) forgetting of SplitCIFAR100 on ER throughout AL rounds.

Figures 4(a) and 4(b) show the performance of ER over the AL rounds at the fifth task on SplitCI-FAR100. Here, AccuACL performs the best, and the performance gap to the AL baselines gets larger as the AL rounds progress. Notably, AccuACL shows very low forgetting, indicating its superiority in preventing catastrophic forgetting. Interestingly, the performances peak around the tenth round of AL. Since the AL was conducted up to 10 rounds for previous tasks, extending beyond 10 rounds for the new task results in an imbalanced data distribution, likely causing the observed performance drop.

## F.3 EFFICACY OF FISHER INFORMATION MATRIX ESTIMATION VIA MEMORY BUFFER

Table 7: Comparison of forgetting and average accuracy between AccuACL and AccuACLFull on SplitCIFAR100($M$=1000) using DER++.

| AL Method | SplitCIFAR100 | |
|---|---|---|
| | $A_{10}(\uparrow)$ | $F_{10}(\downarrow)$ |
| Uniform | 35.9±0.7 | 21.5±0.5 |
| Entropy | 31.7±0.2 | 27.5±0.8 |
| LeastConf | 33.1±1.0 | 27.0±0.6 |
| kCenter | 35.0±0.7 | 24.9±0.5 |
| BADGE | 34.1±2.0 | 27.7±0.7 |
| **AccuACL** | **36.3**±0.4 | 15.0±0.5 |
| **AccuACLFull** | 33.9±0.9 | **14.2**±0.8 |

To investigate the influence of using a memory buffer for estimating the target Fisher information matrix in rehearsal-based CL, we design a modified version of AccuACL, *AccuACLFull*. In contrast

to the original AccuACL, which employs a memory buffer, AccuACLFull leverages all data from previously seen tasks to approximate the target Fisher information matrix. As shown in Table 7, AccuACL achieves competitive results compared to AccuACLFull, indicating that a minimal memory buffer (1000 examples) suffices for adequate estimation. While AccuACLFull reduces forgetting due to more comprehensive parameter estimation, it results in lower average accuracy. We hypothesize that AccuACLFull's extensive data use of the past improves its capacity to retain important parameters from previous tasks that cannot be effectively captured by replaying a memory buffer. However, in a practical scenario of rehearsal-based CL, estimating the Fisher information matrix using only the memory buffer strikes a better balance between retaining past knowledge and integrating new tasks by efficiently selecting new task examples without conflicts with the memory data, instead of attempting to preserve every knowledge of the past.

## G  LIMITATIONS & FUTURE WORKS

While AccuACL has repeatedly shown its efficacy in ACL scenarios, we have yet to validate the suitability of our approach in realistic CL situations, such as noisy labels (Bang et al., 2022; Park et al., 2022), blurry tasks (Bang et al., 2021), and imbalanced data (Chrysakis and Moens, 2020). Moreover, as AccuACL necessitates the use of memory buffer from rehearsal-based CL, a rehearsal-free approach of AccuACL will be a promising approach, possibly with the use of regularization-based CL methods to further accurately maintain the target Fisher information matrix without a memory buffer. Furthermore, as AL for vision-language models (VLMs) (Bang et al., 2024) has shown possibilities, devising an ACL strategy for VLMs to can be a promising research field.

