# OpenReview forum: "Active Learning for Continual Learning: Keeping the Past Alive in the Present"
_ICLR.cc/2025/Conference — ICLR 2025 Poster_

### Official Review · Reviewer_aS7J · 2024-10-24

**Soundness:** 3
**Presentation:** 2
**Contribution:** 3
**Rating:** 6
**Confidence:** 3

**Summary:**

This paper focuses on the scenario that performs Active Learning (AL) in the Continual Learning (CL) process. The paper states that previous AL methods only focus on learning new tasks, which leads to catastrophic forgetting. The authors introduce a novel method called **AccuACL** that improves fisher information-based AL to make a balance between learning new tasks while preserving old knowledge. Empirical results show that **AccuACL** outperforms existing AL methods in standard CL metrics. Overall, this paper opens an important problem and propose a reasonable method to mitigate it.

**Strengths:**

* The problem is important, especially for online deployed models.
* The extensive experiments partly support paper's claims and the effective of the method. Meanwhile, ablation studies help to understand the proposed method.
* The paper provides sufficient details and code in Appendix for readers to reproduce the experiments.

**Weaknesses:**

### Method and theory
1. The proposed method only works for memory-based continual learning methods. It can not be applied on naive continual learning or regularization-based continual learning methods. I understand that memory buffer is necessary for leveraging existing AL method, but I encourage authors to add this as limitations or propose future directions to handle this issue.
2. In Figure 1, it seems like **AccuACL** is good for selecting new samples that related to the learned *features*. However, it is not clear what are those features, and Section 4 does not explain this part clearly either.
3. More background for Active Learning is needed for CL community readers.

### Experiment
1. While theoretical space complexity shows **AccuACL** is more efficient than BAIT, no experiment results are support this claim. An additional experiment for memory usage is needed, like Figure 2(c) for time complexity.
2. In Introduction and Section 5.2, authors claims that:
> "these AL strategies typically pay more attention to new features...", "most AL baselines that most AL baselines that focus only on learning new tasks often perform worse the Uniform."

However, these claims are not confirmed by the experiments. Compared with the Uniform strategy, some existing AL baselines have better forgetting score, but worse average accuracy. This phenomenon might indicates that existing AL baselines are not learning new tasks well either. Learning accuracy metric [1] may be able to support paper's claim. If the claim is true, I expect to see existing AL methods might have higher Learning accuracy. But I am not sure what will the final results be. I encourage authors to do quick additional experiments on CIFAR-10 to see their claims is true or not.

# Reference
[1] Wide Neural Networks Forget Less Catastrophically, ICML 2022

**Questions:**

### Method
1. In Section 4.5, why uses distribution score first and then magnitude score later. Does the order of these two score matters? This is not discussed in Section 5.3 either.
### Future work
1. Does using regularization-based methods can further improve ACL performance? Maybe the authors can consider this idea to further improve ACL.

---

> ### Author Response · Authors · 2024-11-23
> **Author's Response to Reviewer aS7J (1/2)**
>
> > W1-1. The proposed method only works for memory-based continual learning methods. It can not be applied on naive continual learning or regularization-based continual learning methods. I understand that memory buffer is necessary for leveraging existing AL method, but I encourage authors to add this as limitations or propose future directions to handle this issue.
>
> We appreciate the reviewer's insightful comment highlighting the dependency of our method on rehearsal-based CL. We have clarified in the revised draft that AccuACL requires at least minimal access to past distributions. Furthermore, we agree that developing a buffer-free method for estimating the *target* Fisher information matrix is a promising future research direction.
>
> **`See Appendix I of our revised draft.`**
>
> -----
>
>
> > W1-2. In Figure 1, it seems like AccuACL is good for selecting new samples that related to the learned features. However, it is not clear what are those features, and Section 4 does not explain this part clearly either.
>
> We appreciate the reviewer's insightful comment on clarifying the learned features in Figure 1. To clarify, **the learned features in Figure 1 refer to the important parameters learned from past CL tasks**. Intuitively, after identifying important parameters for the past tasks and the new task through the target Fisher information, AccuACL prioritizes sample selection to preserve the parameters important for both the past tasks and the new task.
>
> **`See Section 4.5.1 (lines 356-359) of our revised draft.`**
>
> -----
>
> > W1-3. More background for Active Learning is needed for CL community readers.
>
> Thank you for highlighting this point. We agree that additional background on AL would help bridge the gap for readers more familiar with CL. In response, we have refined the AL part of the related work section (Section 2) by including relevant survey papers [a,b] and providing a clear, straightforward definition of AL. This addition aims to make the paper more accessible to the CL community.
>
> **`See Section 2 (lines 98-100) of our revised draft.`**
>
> [a] A survey of deep active learning, CSUR, 2021
> [b] A survey on active learning: State-of-the-art, practical challenges and research directions, Mathematics, 2023
>
> -----
>
>
> > W2-1. While theoretical space complexity shows AccuACL is more efficient than BAIT, no experiment results are support this claim. An additional experiment for memory usage is needed, like Figure 2c for time complexity.
>
> Thank you for pointing this issue out. We have addressed space complexity concerns by conducting additional experiments analyzing the memory usage of AccuACL. As shown in Figure 2(b) of the revised paper, **AccuACL maintains manageable memory requirements compared to AL baselines**, underscoring its practicality for CL scenarios.
>
> **`See Figure 2(b) of our revised draft.`**
>
> -----
>
> > W2-2. Compared with the Uniform strategy, some existing AL baselines have better forgetting score, but worse average accuracy. This phenomenon might indicates that existing AL baselines are not learning new tasks well either. Learning accuracy metric [1] may be able to support paper's claim. If the claim is true, I expect to see existing AL methods might have higher Learning accuracy. But I am not sure what will the final results be. I encourage authors to do quick additional experiments on CIFAR-10 to see their claims is true or not.
>
>
> We appreciate the reviewer's suggestion to use **learning accuracy** as a metric for plasticity. We conducted experiments on SplitCIFAR100 with DER++, and the results in Table R5 show that AL baselines exhibit higher plasticity (learning accuracy) but lower average accuracy, aligning with our claim that these baselines focus more on learning new tasks.
>
> In contrast, AccuACL explicitly balances the rapid learning of new tasks and the prevention of catastrophic forgetting under a labeling budget, leading to lower learning accuracy but superior performance on forgetting. AccuACL's superior **average** accuracy emphasizes the **importance of maintaining this balance for effective ACL performance**.
>
> **Table R5**: Learning accuracy($LA_{10}$), forgetting($F_{10}$), and average accuracy($A_{10}$) comparison between AL baselines and AccuACL on SplitCIFAR100($M$=500) on DER++. Standard deviation is presented in parentheses.
> |           |   Uniform   |   Entropy   | LeastConf |   kCenter   |   BADGE   |  AccuACL  |
> |:-------:|:-------:|:-------:|:---------:|:-------:|:-----:|:---------:|
> | $LA_{10}$ |  62.2(0.6)  |  64.9(0.2)  | 65.8(1.3) |  65.0(0.3)  | **65.9**(0.5) | 56.6(0.3) |
> |  $F_{10}$ | 38.7(1.3)   |  46.9(0.6)  | 48.9(1.1) |  43.4(1.1)  |   45.6(1.0)   | **29.3**(0.1) |
> |  $A_{10}$ | 27.4(0.6)   |  22.6(0.4)  | 21.7(0.3) |  25.9(0.0)  |   24.8(0.4)   | **30.0**(0.4) |
>
> **`See Appendix E.4 of our revised draft.`**
>
> -----

---

> ### Author Response · Authors · 2024-11-23
> **Author's Response to Reviewer aS7J (2/2)**
>
> > Q1. In Section 4.5, why uses distribution score first and then magnitude score later. Does the order of these two score matters? This is not discussed in Section 5.3 either.
>
> As mentioned in lines 454–457, the magnitude score $\mathcal{M}(\cdot)$ focuses only on the size of the information, thereby emphasizing the quick learning of new tasks. In contrast, the distribution score $\mathcal{D}(\cdot)$ focuses on the examples whose information distribution is closest to the target Fisher information. In summary, **employing $\mathcal{D}(\cdot)$ as the primary criterion allows for an AL algorithm that is more aligned with the motivation of our paper**, which is to maintain a balance between the two learning properties of ACL: prevention of catastrophic forgetting and quick learning of new tasks. Ensuring the magnitude of informativeness after establishing the balance has shown effectiveness through empirical evaluation.
>
> **`See Section 5.3 (lines 482-485) of our revised draft.`**
>
> -----
>
> > Q2. Does using regularization-based methods can further improve ACL performance? Maybe the authors can consider this idea to further improve ACL.
>
> Thank you for the insightful question. As AccuACL leverages the Fisher information matrix, we share your interest in combining it with regularization-based CL methods, which also utilize the Fisher information matrix. If regularization-based methods can effectively preserve the values of important parameters from past tasks, this integration could potentially lead to a rehearsal-free version of AccuACL. We find this topic to be an exciting avenue for future work and appreciate your input in highlighting this possibility.
>
> **`See Appendix I of our revised draft.`**
>
> -----

---

> > ### Comment · Reviewer_aS7J · 2024-11-25
> >
> > I thank the authors for the detailed response and additional experiments. All of my questions are answered. Overall, I support the acceptance of this paper.

---

> > > ### Author Response · Authors · 2024-11-25
> > >
> > > Once more, we are profoundly grateful for your invaluable feedback and support for acceptance.

---

### Official Review · Reviewer_fGgu · 2024-10-26

**Soundness:** 3
**Presentation:** 3
**Contribution:** 3
**Rating:** 8
**Confidence:** 4

**Summary:**

The paper focuses on active continual learning (ACL) and proposes AccuACL (Accumulated Informativeness-based Active Continual Learning) to effectively balance the prevention of catastrophic forgetting and the rapid learning of new tasks. This method proposes the accumulated informativeness modeled by the Fisher information matrix, which is further approximated by the diagonal Fisher information embedding to reduce the calculation cost. Experimental results show that AccuACL outperforms existing active learning baselines across various continual learning algorithms.

**Strengths:**

1. The motivation is clear and well-justified, aiming to enhance previous methods by addressing both forgetting and rapid learning in subset selection for training.

2. The proposed method is based on the two theorems, making the algorithm be more confident and efficient.

3. Experimental results demonstrate the strength of the proposed method, significantly improving accuracy while reducing forgetting.

4. The paper is well-organized and most components are clear and easy to understand.

**Weaknesses:**

**Major**

1. The method employs buffer data to evaluate the current parameter importance with respect to the historical task. However, a small buffer can not adequately represent the complete data distribution of historical tasks, which is easy to overfit. Then the network may memorize these buffer samples and exhibit low parameter sensitivity, leading to an incorrect Fisher information matrix.

2. The paper focuses solely on the Fisher information matrix of linear classifiers. However, linear classifiers in CL often exhibit task-recency bias, making them prone to favoring new classes [1,2,3]. This raises concerns about the accuracy of the Fisher information matrix calculation. How can an unbiased Fisher information matrix be derived from a biased linear classifier?

3. The description of the greedy algorithm for subset selection is unclear (line 6 in Algorithm 1 requires further explanation). My understanding is as follows:

   a) Randomly sample a subset and calculate the Fisher information embedding according to Theorem 4.2.

   b) Calculate the magnitude score and distribution score of the current subset.

   c) Randomly sample a lot of subsets and repeat steps a) and b). After that, comparing scores across these subsets, and then, selecting the subset with the highest score as the final selection subset $X$.

4. If my understanding is correct, how to guarantee the convergence of the greedy algorithm and ensure that it can choose the optimal subset? Additionally, detailed settings should be provided, such as the subset sampling method, whether subsets intersect, and the size of the subsets.

5. Regarding the budget experiment. There is no clear definition of budget and no experiments pointing out the budget limitation.


**Minor**

1. In Table 3 (lines 472-474), how to incorporate historical knowledge into the greedy algorithm? Why use the memory buffer as the initial subset?

[1] Supervised contrastive replay: Revisiting the nearest class mean classifier in online class-incremental continual learning. CVPR, 2021.

[2] Cba: Improving online continual learning via continual bias adaptor. ICCV, 2023.

[3] Overcoming recency bias of normalization statistics in continual learning: Balance and adaptation. NIPS, 2024.

If all my concerns are addressed, I'll gladly increase the score.

**Questions:**

1. In active learning, why is unlabeled data not utilized in an unsupervised manner to further enhance the training process?

2. In Figure 3, the choice of lambda depends on the amount of unlabeled data. Does this conclusion also hold in imbalanced datasets?

---

> ### Author Response · Authors · 2024-11-23
> **Author's Response to Reviewer fGgu (1/3)**
>
> > W1. The method employs buffer data to evaluate the current parameter importance with respect to the historical task. However, a small buffer can not adequately represent the complete data distribution of historical tasks, which is easy to overfit. Then the network may memorize these buffer samples and exhibit low parameter sensitivity, leading to an incorrect Fisher information matrix.
>
> We appreciate the reviewer's concern about the potential for incorrect Fisher information matrix estimation with a small memory buffer. To address this issue, we developed a variant, AccuACLFull, that uses all past data for Fisher matrix estimation. As shown in Table R4, AccuACL achieves competitive results compared to AccuACLFull, indicating that a **minimal memory buffer(500 examples) suffices for adequate estimation**. While AccuACLFull reduces forgetting due to more comprehensive parameter estimation, it results in lower average accuracy.
>
> We hypothesize that in a practical scenario of rehearsal-based CL, estimating the Fisher information matrix using only the memory buffer strikes a better balance between retaining past knowledge and integrating new tasks by efficiently **selecting new task samples without conflicts with the memory data**, instead of attempting to preserve every knowledge of the past.
>
>
> ***Table R4***: Comparison of forgetting and average accuracy between AccuACL and AccuACLFull on SplitCIFAR100($M$=500) on DER++. Standard deviation is presented in parentheses.
> |      AL     | $A_{10}$              | $F_{10}$              |
> |-------------|-----------------------|-----------------------|
> | Uniform     | 35.9 (0.7) | 21.5 (0.5)        |
> | Entropy     | 31.7 (0.2)       | 27.5 (0.8)        |
> | LeastConf   | 33.1 (1.0)       | 27.0 (0.6)        |
> | kCenter     | 35.0 (0.7)       | 24.9 (0.5)        |
> | BADGE       | 34.1 (2.0)       | 27.7 (0.7)        |
> | AccuACL     | **36.3** (0.4)    | 15.0 (0.5) |
> | AccuACLFull | 33.9 (0.9)      | **14.2** (0.8)    |
>
> **`See Appendix E.3 of our revised draft.`**
>
> -----
>
> > W2. The paper focuses solely on the Fisher information matrix of linear classifiers. However, linear classifiers in CL often exhibit task-recency bias, making them prone to favoring new classes [1,2,3]. This raises concerns about the accuracy of the Fisher information matrix calculation. How can an unbiased Fisher information matrix be derived from a biased linear classifier?
>
> We appreciate the reviewer highlighting concerns about task-recency bias and its impact on Fisher information matrix estimation. We gently argue that **AccuACL's query selection method helps alleviate this bias by prioritizing examples that preserve critical parameters of past tasks**, thereby reducing the influence of recency bias in linear classifiers as well as leading to a more reliable Fisher information matrix calculation for the upcoming AL rounds.
>
> As shown in Table R5, AccuACL achieves balanced performance between plasticity (measured by learning accuracy [a], $LA_{10}$) and stability (measured by forgetting, $F_{10}$). This balance leads to superior overall accuracy (measured by average accuracy, $A_{10}$), demonstrating AccuACL's ability to mitigate task-recency bias effectively.
>
> **Table *R5***: Learning accuracy($LA_{10}$), forgetting($F_{10}$), and average accuracy($A_{10}$) comparison between AL baselines and AccuACL on SplitCIFAR100($M$=500) on DER++. Standard deviation is presented in parentheses.
> |           |   Uniform   |   Entropy   | LeastConf |   kCenter   |   BADGE   |  AccuACL  |
> |:-------:|:-------:|:-------:|:---------:|:-------:|:-----:|:---------:|
> | $LA_{10}$ |  62.2(0.6)  |  64.9(0.2)  | 65.8(1.3) |  65.0(0.3)  | **65.9**(0.5) | 56.6(0.3) |
> |  $F_{10}$ | 38.7(1.3)   |  46.9(0.6)  | 48.9(1.1) |  43.4(1.1)  |   45.6(1.0)   | **29.3**(0.1) |
> |  $A_{10}$ | 27.4(0.6)   |  22.6(0.4)  | 21.7(0.3) |  25.9(0.0)  |   24.8(0.4)   | **30.0**(0.4) |
>
> **`See Appendix E.4 of our revised draft.`**
>
> [a] Wide neural networks forget less catastrophically, ICML, 2022
>
> -----

---

> ### Author Response · Authors · 2024-11-23
> **Author's Response to Reviewer fGgu (2/3)**
>
> > W3. The description of the greedy algorithm for subset selection is unclear.
>
> > W4. Additionally, detailed settings should be provided, such as the subset sampling method, whether subsets intersect, and the size of the subsets.
>
> We appreciate the reviewer’s insightful feedback. Rather than randomly selecting subsets multiple times, our approach employs an example-wise scoring method using the distribution score and magnitude score to rank all samples. We then select the top-k examples based on these criteria. We have clarified the two-stage sampling process in **Algorithm 1** in our revised draft.
>
> When a new task(unlabeled data pool $U_t$) is introduced:
> 1. Randomly sample a subset from $U_t$, label them, and train the model from $\theta_{t-1}$ (the model checkpoint from task $t-1$), to obtain $\theta_t$.
> 2. Using $\theta_t$, we calculate the Fisher information embedding for $U_t$ according to Theorem 4.2 and calculate the **target** Fisher information embedding $\mathbf{F}_t$.
> 3. Over-sample(twice the budget $b$) examples whose Fisher information embedding's distribution is distributionally closest to $\mathbf{F}_t$.
> 4. Narrow down to $b$ samples that have the highest Fisher information magnitude.
> 5. Acquire labels for the $b$ samples, then train a better model estimate $\theta_t$ from $\theta_{t-1}$, and remove the labeled examples from $U_t$.
> 6. Repeat 2-5.
>
> **`See Algorithm 1 of our revised draft.`**
>
> -----
>
> > W4. If my understanding is correct, how to guarantee the convergence of the greedy algorithm and ensure that it can choose the optimal subset?
>
> Regarding convergence guarantees, we clarify that the objective function in Equation 8 is not submodular, meaning that the greedy algorithm does not offer a bounded approximation guarantee. To address this issue, we introduced two Fisher-optimality-preserving properties under complementary assumptions, as detailed in Section 4.4 and Theorem 4.3. These properties provide a theoretical foundation for the utility of our approach, even without convergence guarantees.
>
> Moreover, we acknowledge that the **intuition of the distribution score $\mathcal{D}(\cdot)$ aligns well with our motivation to balance quick adaptation to new tasks and prevention of catastrophic forgetting**, as it prioritizes the samples whose information distribution matches the target Fisher information matrix, therefore leading to superior ACL performance.
>
> -----
>
>
> > W5. Regarding the budget experiment. There is no clear definition of budget and no experiments pointing out the budget limitation.
>
> Thank you for pointing this presentation issue out. The budget $b$ in Algorithm 1 refers to the per-round labeling budget, set as {100, 200, 300} for SplitCIFAR10, SplitCIFAR100, and SplitTinyImageNet, respectively, resulting in total per-task labeling budgets of {1000, 2000, 3000}.
>
> Regarding budget limitations, we conducted experiments analyzing the performance trends for varying per-task labeling budgets, as shown in Table R1. Overall, **AccuACL consistently outperforms AL baselines with increasing query numbers**, highlighting its robustness and adaptability.
>
> **Table *R1***: Average accuracy for SplitCIFAR100($M$=500) on ER for different labeling budgets per task. Standard deviation is presented in parentheses.
> | Per-task budget        | 100   | 500   | 1000   | 1500   | 2000   | 2500   | 3000   |
> |:----------|:---------------|:---------------|:----------------|:----------------|:----------------|:----------------|:----------------|
> | Uniform   | **6.2**(0.3)       | 10.8(0.3)      | 10.9(0.3)       | 10.7(0.1)       | 11.0(0.4)       | 11.0(0.3)       | 11.6(0.5)       |
> | Entropy   | 4.8(0.3)       | 7.3(0.7)       | 7.6(0.1)        | 8.0(0.5)        | 8.5(0.3)        | 8.8(0.4)        | 9.6(0.3)        |
> | LeastConf | 4.7(0.6)       | 7.7(0.7)       | 7.4(0.3)        | 7.9(0.2)        | 8.8(0.1)        | 8.9(0.1)        | 9.3(0.4)        |
> | kCenter   | 4.6(0.3)       | 9.0(0.4)       | 9.2(0.2)        | 9.6(0.6)        | 9.7(0.7)        | 10.3(0.5)       | 10.4(0.0)       |
> | BADGE     | 5.6(0.1)       | 9.5(0.7)       | 8.8(0.1)        | 9.1(0.5)        | 9.0(0.0)        | 9.4(0.4)        | 9.6(0.0)        |
> | AccuACL   | 5.5(0.4)       | **12.2**(0.6)      | **14.4**(1.5)       | **15.1**(0.8)       | **14.5**(0.6)       | **15.0**(0.3)       | **13.4**(0.6)       |
>
> **`See Figure 2(a) of our revised draft.`**
>
> -----

---

> ### Author Response · Authors · 2024-11-23
> **Author's Response to Reviewer fGgu (3/3)**
>
> > Minor W1. In Table 3 (lines 472-474), how to incorporate historical knowledge into the greedy algorithm? Why use the memory buffer as the initial subset?
>
> We appreciate the reviewer's insightful question. As iterative algorithms, such as BADGE and KCenterGreedy, allow a custom initial subset, we initialize the subset with the memory buffer to incorporate past knowledge into these AL baselines. However, as shown in Table 3, **adding past information to these AL baselines even deteriorates ACL performance**. We hypothesize that these methods, which prioritize geometric diversity, shift focus to selecting new task samples significantly different from the memory samples, thereby further emphasizing quick learning of new tasks. This result highlights the importance of careful incorporation of past knowledge into AL baselies and verifies **AccuACL's effectiveness in appropriately integrating past knowledge into the query strategy**.
>
>
> -----
>
> > Q1. In active learning, why is unlabeled data not utilized in an unsupervised manner to further enhance the training process?
>
> We appreciate the reviewer's suggestion to clarify the use of unlabeled data in an unsupervised manner. Active learning (AL) and semi-supervised learning (SSL) address annotation costs differently: AL focuses on **selecting** the most informative unlabeled data examples to maximize information gain, whereas SSL **exploits** partially labeled datasets to improve generalization. Thus, AL emphasizes querying unlabeled data rather than directly leveraging unlabeled data for optimization.
>
> That said, the integration of AL and SSL has led to the emerging field of active semi-supervised learning [a], which combines the strengths of both to efficiently utilize labeled and unlabeled data. However, this direction is beyond the scope of this work.
>
> [a] Semantic segmentation with active semi-supervised learning, WACV, 2023
>
> -----
>
> > Q2. In Figure 3, the choice of lambda depends on the amount of unlabeled data. Does this conclusion also hold in imbalanced datasets?
>
> Thank you for raising this important point regarding the applicability of $\lambda$. We agree that evaluating our approach in the context of imbalanced datasets is essential for a more comprehensive analysis. Beyond our current formulation, $\lambda$ could potentially be estimated more effectively by considering the number of previously seen tasks or by adapting it based on the inter-distribution distance between past tasks and the new task. These adaptive strategies represent promising directions for future research, and we appreciate your suggestion for highlighting this important topic.
>
> -----

---

> ### Author Response · Authors · 2024-11-27
> **Gentle Reminder**
>
> We greatly appreciate your insightful feedback and genuinely hope our response has resolved your concerns. Should any concerns persist, we are willing to engage in discussion with you throughout the remainder of the discussion period.

---

> ### Comment · Reviewer_fGgu · 2024-11-28
> **Thanks for the response.**
>
> Thank you for the detailed and clear response. Most of my concerns have been solved and I am willing to raise my score to 8.

---

> > ### Author Response · Authors · 2024-11-28
> >
> > We greatly appreciate your invaluable feedback and support for acceptance. Thank you once again for your thoughtful and constructive comments.

---

### Official Review · Reviewer_Fzgq · 2024-11-02

**Soundness:** 2
**Presentation:** 2
**Contribution:** 2
**Rating:** 3
**Confidence:** 4

**Summary:**

The paper proposes an active learning method for continual learning that balances the prevention of catastrophic forgetting and the ability to quickly learn new tasks.

**Strengths:**

Studying active learning in the continual learning setting is interesting.

**Weaknesses:**

1.	The novelty of the paper is somewhat limited, as the use of the Fisher information matrix and replay mechanisms are already well-known methods in continual learning.

2.	The citations in the related work section are outdated, giving the impression that the authors may not be fully aware of the latest advancements in continual learning, particularly beyond replay and regularization-based methods. Please see the survey [1] and the related work section of [2] and [3].

3.	It would be beneficial to examine how the system and baseline models perform as the number of queries increases.

4.	The baseline methods used in this paper are also outdated. See those in [2] and [3].

5.	The current accuracy results are quite low and significantly fall short of existing state-of-the-art methods.

6.	Given that your method selects only a small number of samples, it would be essential to compare it with recent few-shot continual learning approaches [4], [5], [6], and [7].

7.	Catastrophic forgetting does not seem to be the main problem anymore. Inter-task class discrimination may pose a greater challenge. Please refer to [2] and [3].

8.	The results on the full dataset are notably low too, and it is unclear what method was used in the “full” setting. Please see the papers mentioned above for comparison.

9.     The paper is overly formal and not easy to follow. Some more descriptions of intuitions will be helpful.

10.   The space completely is very high for continual learning.

[1] Wang et al. A comprehensive survey of continual learning: theory, method and application. IEEE Transactions on Pattern Analysis and Machine Intelligence. 2024.

[2] Lin et al., "Class Incremental Learning via Likelihood Ratio-Based Task Prediction," ICLR 2024.

[3] Wang et al. BEEF: Bi-compatible class-incremental learning via energy-based expansion and fusion. ICLR 2023.

[4] Mazumder et al. Few-Shot Lifelong Learning. AAAI-2021

[5] Zhou et al. Forward compatible few-shot class-incremental learning. CVPR- 2022.

[6] Song et al. Learning with fantasy: Semantic-aware virtual contrastive constraint for few-shot class-incremental learning. CVPR-2023

[7] Tian et al. A survey on few-shot class-incremental learning. Neural Networks. 2024.

**Questions:**

See the previous sections

---

> ### Author Response · Authors · 2024-11-23
> **Author's Response to Reviewer Fzgq (1/4)**
>
> > W1. The novelty of the paper is somewhat limited, as the use of the Fisher information matrix and replay mechanisms are already well-known methods in continual learning.
>
> Thank you for your comment. We respectfully contend that **the paper's novelty lies in its novel ACL algorithm, which effectively balances the prevention of catastrophic forgetting with the capacity to rapidly acquire new tasks**, through the Fisher Information Matrix (FIM) and a replay mechanism. We are of the opinion that the novelty of our work would not be compromised just by the inclusion of the FIM and the replay mechanism, which are frequently employed in CL. In detail, our novelty is elucidated as follows:
>
> 1. ***New* Usage of the FIM.** While existing CL works use the FIM as a **regularizer** to retain information from past tasks when learning new tasks [a, b], we use it as a **metric for sample selection** to assess the informativeness of each unlabeled sample in the ACL scenario. Our proposed sample selection approach, AccuACL, precisely examines the FIM to preserve the quantity of the information in a selected subset, which is **not** considered in the existing CL works. This work demonstrates a new use case of the FIM in the context of ACL.
> 2. **Theoretical Justification.** We further provide theoretical justification of our sample selection approach in terms of utilizing the FIM effectively. Specifically, we introduce **two Fisher-optimality-preserving properties under complementary assumptions**, which lead to a novel scoring criterion that ensures the optimality of the selected subset, as detailed in Section 4.4 and Theorem 4.3.
>
> Overall, we sincerely hope that our response has addressed your concern regarding novelty.
>
> **`See lines 19-22 of our revised draft.`**
>
> [a] Overcoming catastrophic forgetting in neural networks, PNAS, 2017
>
> [b] Memory aware synapses: Learning what (not) to forget, ECCV, 2018
>
> -----
>
> > W2. The citations in the related work section are outdated, giving the impression that the authors may not be fully aware of the latest advancements in continual learning, particularly beyond replay and regularization-based methods. Please see the survey [1] and the related work section of [2] and [3].
>
> Thank you for informing us of recent relevant studies. To ensure that our related work section reflects the latest advancements in CL, we have expanded it by including additional citations: [c, d, e] for rehearsal-based CL and [f, g, h] for dynamic structure-based CL. This update aims to provide a more comprehensive overview of the field while maintaining conciseness.
>
> **`See Section 2, lines 110-112 and 115-116 of our revised draft.`**
>
> [c] Loss Decoupling for Task-Agnostic Continual Learning, NIPS, 2023
>
> [d] Learnability and algorithm for continual learning, ICML, 2023
>
> [e] Class incremental learning via likelihood ratio based task prediction, ICLR, 2024
>
> [f] Der: Dynamically expandable representation for class incremental learning, CVPR, 2021
>
> [g] A Model or 603 Exemplars: Towards Memory-Efficient Class-Incremental Learning, ICLR, 2023
>
> [h] Beef: Bi-compatible class-incremental learning via energy-based expansion and fusion, ICLR, 2023
>
>
> -----
>
> > W3. It would be beneficial to examine how the system and baseline models perform as the number of queries increases.
>
> We have conducted additional experiments on SplitCIFAR100, examining the average accuracy trend as the number of per-task queried examples increases (see Table R1 below). The results show that **AccuACL consistently outperforms AL baselines across varying query numbers**, demonstrating its robustness and adaptability.
>
> ***Table R1***: Average accuracy of SplitCIFAR100($M$=500) on ER for different labeling budgets per task. Standard deviation is presented in parentheses.
> | Per-task budget        | 100   | 500   | 1000   | 1500   | 2000   | 2500   | 3000   |
> |:----------|:---------------|:---------------|:----------------|:----------------|:----------------|:----------------|:----------------|
> | Uniform   | **6.2**(0.3)       | 10.8(0.3)      | 10.9(0.3)       | 10.7(0.1)       | 11.0(0.4)       | 11.0(0.3)       | 11.6(0.5)       |
> | Entropy   | 4.8(0.3)       | 7.3(0.7)       | 7.6(0.1)        | 8.0(0.5)        | 8.5(0.3)        | 8.8(0.4)        | 9.6(0.3)        |
> | LeastConf | 4.7(0.6)       | 7.7(0.7)       | 7.4(0.3)        | 7.9(0.2)        | 8.8(0.1)        | 8.9(0.1)        | 9.3(0.4)        |
> | kCenter   | 4.6(0.3)       | 9.0(0.4)       | 9.2(0.2)        | 9.6(0.6)        | 9.7(0.7)        | 10.3(0.5)       | 10.4(0.0)       |
> | BADGE     | 5.6(0.1)       | 9.5(0.7)       | 8.8(0.1)        | 9.1(0.5)        | 9.0(0.0)        | 9.4(0.4)        | 9.6(0.0)        |
> | AccuACL   | 5.5(0.4)       | **12.2**(0.6)      | **14.4**(1.5)       | **15.1**(0.8)       | **14.5**(0.6)       | **15.0**(0.3)       | **13.4**(0.6)       |
>
> **`See Figure 2(a) of our revised draft.`**
>
> -----

---

> ### Author Response · Authors · 2024-11-25
> **Author's Response to Reviewer Fzgq (2/4)**
>
> > W4. The baseline methods used in this paper are also outdated. See those in [2] and [3].
>
> Thank you again for your comment. First of all, we would like to clarify that **AccuACL is agnostic to rehearsal-based CL methods**. Consequently, we concentrated primarily on established CL methods rather than on emerging CL techniques. We concur that recent CL methods should be evaluated as well. Due to the short rebuttal period, we could incorporate only one more baseline. In conjunction with **`W7`**, we opted to incorporate a recent rehearsal-based CL method, LODE [c], due to its capability to manage inter-task class discrimination. As shown in Table R3, **AccuACL consistently outperforms the AL baselines across the majority of settings together with a new CL method**. Entropy and LeastConf exhibited low forgetting on SplitTinyImageNet, attributable to the exceedingly low average accuracy. In other words, there is less to forget when learning is minimal.
>
> ***Table R3***: Perforance comparison of AL baselines and **AccuACL** combined with **LODE [c]** on SplitCIFAR10, SplitCIFAR100, and SplitTinyImageNet.
> |            | SplitCIFAR10 |               | SplitCIFAR100 |               | SplitTinyImageNet |               |
> |------------|---------------|---------------|---------------|---------------|-------------------|---------------|
> | AL Methods | $A_{5}$      | $F_{5}$      | $A_{10}$      | $F_{10}$      | $A_{10}$          | $F_{10}$      |
> | Uniform    | 37.6(0.8)     | 25.2(5.9)     | 27.4(1.0)     | 28.4(1.2)     | 11.8(0.8)         | 23.5(0.3)     |
> | Entropy    | 36.2(2.1)     | 25.1(1.4)     | 25.9(0.9)      | 28.6(0.7)     | 10.3(0.4)         | **21.8**(0.4) |
> | LeastConf  | 34.7(1.0)     | 26.6(3.6)     | 26.4(0.9)     | 28.8(0.9)     | 9.9(0.2)          | 21.9(0.5)     |
> | kCenter    | 33.9(1.0)     | 30.1(1.9)     | 26.7(0.9)     | 30.5(0.4)     | 11.0(0.4)         | 26.5(1.0)     |
> | BADGE      | **38.0**(0.4) | 29.0(1.0)     | 26.6(0.5)     | 29.0(0.3)     | 11.2(0.2)         | 24.2(0.6)     |
> | **AccuACL**    | 36.4(1.0)     | **24.2**(1.0) | **28.4**(0.5) | **26.4**(0.7) | **12.4**(0.9)     | 23.5(0.6)     |
>
> Moreover, we will incorporate additional recent rehearsal-based CL baselines in the camera-ready version.
>
> **`See Appendix E.1 of our revised draft.`**
>
> -----
>
> > W5. The current accuracy results are quite low and significantly fall short of existing state-of-the-art methods.
>
> > W8. The results on the full dataset are notably low too, and it is unclear what method was used in the "full" setting. Please see the papers mentioned above for comparison.
>
> We greatly appreciate your meticulous review of the experimental results.
>
> **`W5:`** A number of configurations for assessing CL methods are prevalent in the literature. That is, to the best of our knowledge, there is no definitive standard configuration for CL works. The experimental configurations in [2, 3] diverge from ours in terms of buffer size, network architecture, number of epochs and the optimization method. The distinction between [2] and ours can be summarized in Table R4.
>
> **Table R4**: Comparison of the experimental setups for CL between BEEF[2] and AccuACL.
> | Aspect  	| Model Architecture 	| Learning Rate 	| Optimizer 	| Epochs 	| Batch Size 	| Memory Size 	|
> | ----------|---------------------------|----------------------------|----------------------|--------------|----------------|---------------------|
> | BEEF[2] 	| ResNet-32          	| 0.1           	| SGD       	| 170    	| 128        	| 2000        	|
> | AccuACL 	| ResNet-18          	| 0.01          	| SGD       	| 20     	| 16         	| 500/1000    	|
>
> **We follow the Avalanche codebase [i], with similar performance levels reported in [j]**. Please refer to Table 2 of [j], which shows comparable ranges to the values presented in our paper.
>
> **`W8:`** The ***full*** setting refers to the fully-supervised setting, identical to the conventional CL problem, where all training examples are labeled. **`W8`** is addressed by the response to **`W5`**.
>
> Overall, we have double-checked that **our experiment results are correct**. We sincerely hope that our response has addressed your concern regarding the validity of our evaluation.
>
> **`See line 415 of our revised draft.`**
>
> [i] Avalanche: an end-to-end library for continual learning, CVPR Workshop, 2021
>
> [j] Cascaded Scaling Classifier: class incremental learning with probability scaling, ArXiv, 2024
>
> -----

---

> > ### Comment · Reviewer_Fzgq · 2024-11-25
> > **Regarding the Rebuttal**
> >
> > Thank you for your response. Unfortunately, I find most of the explanations provided unsatisfactory. Leaving other factors aside, the accuracy results fall significantly short compared to state-of-the-art methods. Without competitive accuracy, the practical utility of the approach is limited. Moreover, there are existing methods that effectively address catastrophic forgetting. Regarding space complexity, while d may be constant, the other components are not, as continual learning involves ongoing learning without end. Given these concerns, I would like to retain my current score.

---

> > > ### Author Response · Authors · 2024-11-26
> > > **Additional Experimental Results**
> > >
> > > To address your concerns, we quickly conducted additional experiments aligned with the configuration in Table 2 of [2] (though not perfectly aligned due to time constraints). **Simply by changing the experiment configuration, $A_{10}$ of AccuACL has been increased from 22.0 to 39.4**, and the superiority of AccuACL is properly preserved. Thus, we contend that the absolute value of the accuracy can be improved easily to some extent by CL configurations. We are changing the codebase so that it is aligned more with [2], expecting that the absolute value of the accuracy will increase further. **We will continue to provide updates on additional experiment results with enhanced accuracy**.
> > >
> > > The results in Table A1 demonstrate that AccuACL achieves superior ACL performance compared to the second-best AL algorithm, Uniform (based on Table 1 of our draft). Furthermore, AccuACL outperforms the full version of ACL (equivalent to the conventional CL setting), consistent with our findings in Table 1, where AccuACL + ER on SplitCIFAR100 exceeds the full performance. These results validate the correctness of our algorithm.
> > >
> > > ***Table A1***: Comparison of forgetting and average accuracy on SplitCIFAR100($M$=2000) on ER.
> > > |   |$A_{10}$|$F_{10}$|
> > > |----|----|----|
> > > | Full    	   | 36.1     	| 53.6     	|
> > > | Uniform    | 35.5     	| 56.8    	|
> > > | AccuACL  | **39.4**	|  **34.2** 	|
> > >
> > > We also emphasize that our focus is on selecting informative unlabeled data for labeling. Using formally defined criteria, accumulated informativeness, we theoretically assess the informativeness of each data point from two aspects, enabling a balanced integration of these aspects to suit ACL scenarios. We believe this theoretical foundation is a valuable contribution that can benefit many CL approaches, as **our work is the first to assess unlabeled data in terms of its potential to prevent catastrophic forgetting**—an essential aspect of CL.
> > >
> > > We appreciate your concerns and welcome further discussions to refine and improve our work.

---

> ### Author Response · Authors · 2024-11-25
> **Author's Response to Reviewer Fzgq (3/4)**
>
> > W6. Given that your method selects only a small number of samples, it would be essential to compare it with recent few-shot continual learning approaches [4], [5], [6], and [7].
>
> This is a really great idea. Active continual learning (ACL) concentrates on **selecting** the most informative unlabeled data for labeling, whereas few-shot continual learning (FSCL) concentrates on **exploiting** partially labeled data, particularly when the number of labeled examples is extremely small. ACL typically allocates a greater labeling budget than FSCL. Thus, we believe that the utilization of FSCL for optimization during the initial AL rounds could enhance model estimation, thereby facilitating more effective subsequent querying. We will reserve this intriguing concept for future research.
>
> -----
>
> > W7. Catastrophic forgetting does not seem to be the main problem anymore. Inter-task class discrimination may pose a greater challenge. Please refer to [2] and [3].
>
> While we do agree that inter-task class discrimination is an important challenge in recent CL research, we believe that **mitigating catastrophic forgetting remains fundamental**, as it is an inherent challenge in any CL setting. In the most recent top-tier conferences, there are approximately 12 (out of 17 CL)  publications in ICLR 2024 and 9 (out of 18 CL) publications in ICML 2024 that address catastrophic forgetting in their abstract, indicating that it continues to be a fundamental challenge.
>
> Additionally, the aforementioned methods in [2, 3] also face catastrophic forgetting when the model's capacity to accommodate new knowledge is exceeded. We believe that developing an ACL strategy capable of distinguishing unlabeled samples that help prevent forgetting can significantly contribute to the research community. **Our theoretical foundation is expected to provide insights that may lead to further applications**, particularly beyond class-incremental learning scenarios.
>
> Nevertheless, we also acknowledge the importance of inter-task discrimination in class-incremental learning. To demonstrate the relevance of AccuACL in CL strategies that address the inter-task discrimation, we have conducted additional experiments using LODE [c], which decouples intra-task and inter-task discrimination with a hyperparameter $\rho$ to control inter-task discrimination strength. As shown in Table R5, **AccuACL consistently outperforms other AL baselines across diverse $\rho$ values**, **demonstrating its robustness across varying levels of inter-task discrimination**.
>
> **Table R5**: Performance comparison of AccuACL and baseline methods on SplitCIFAR100($M$=500) using LODE across different values of $\rho$. Standard deviation is presented in parentheses.
> | $\rho$        | 0.1   |    | 0.2   |    | 0.3   |    | 0.4   |    | 0.5   |    |
> |:----------|:---------------------|:----------------------|:---------------------|:----------------------|:---------------------|:----------------------|:---------------------|:----------------------|:---------------------|:----------------------|
> | Active Learning        | $A_{10}$   | $F_{10}$   | $A_{10}$   | $F_{10}$   | $A_{10}$   | $F_{10}$   | $A_{10}$   | $F_{10}$   | $A_{10}$   | $F_{10}$   |
> | Uniform   | 27.9(0.8)            | 21.6(1.0)             | 27.4(1.0)            | 28.4(1.2)             | 27.0(0.6)            | 31.9(1.4)             | 26.3(1.0)            | 34.3(1.3)             | 25.9(0.8)            | 35.8(1.3)             |
> | Entropy   | 26.3(0.5)            | 21.0(0.7)             | 25.9(0.9)            | 28.6(0.7)             | 25.4(0.8)            | 32.0(0.6)             | 24.9(1.0)            | 34.8(0.0)             | 24.0(0.5)            | 36.9(0.8)             |
> | LeastConf | 26.9(1.2)            | 21.2(2.3)             | 26.4(0.9)            | 28.8(0.9)             | 25.4(1.0)            | 33.4(0.9)             | 24.5(0.3)            | 36.2(1.6)             | 24.6(1.4)            | 37.5(1.4)             |
> | kCenter   | 26.7(0.7)            | 24.4(0.5)             | 26.7(0.9)            | 30.5(0.4)             | 25.6(1.2)            | 34.7(0.8)             | 25.5(0.7)            | 36.9(0.7)             | 24.3(0.5)            | 39.1(0.2)             |
> | BADGE     | 27.1(0.3)            | 21.9(0.5)             | 26.6(0.5)            | 29.0(0.3)             | 25.4(0.2)            | 33.4(0.5)             | 24.7(0.6)            | 36.2(0.7)             | 25.1(0.5)            | 36.5(1.1)             |
> | AccuACL   | **29.1**(0.4)            | **20.3**(1.1)             | **28.4**(0.5)            | **26.4**(0.7)             | **27.4**(0.6)            | **29.8**(0.4)             | **27.4**(0.8)            | **31.0**(0.6)             | **27.3**(0.5)            | **32.6**(0.8)             |
>
> **`See Appendix E.1 of our revised draft.`**
>
> -----

---

> ### Author Response · Authors · 2024-11-25
> **Author's Response to Reviewer Fzgq (4/4)**
>
> > W9. The paper is overly formal and not easy to follow. Some more descriptions of intuitions will be helpful.
>
> Your feedback is greatly appreciated. We acknowledge that the underlying intuitions could be more clearly explained in certain sections. We will undoubtedly continue to refine the paper's presentation.
>
> -----
>
> > W10. The space completely is very high for continual learning.
>
> AccuACL has a space complexity of $O((m+n)dK)$, where $m$ is the memory buffer size, $n$ is the data pool size, $d$ is the embedding dimensionality, and $K$ is the total number of classes. As $m$, $d$, and $K$ can be regarded as constant numbers, the space complexity is **linear** to the number of unlabeled examples.
>
> To further verify our claim, we have conducted additional experiments analyzing the memory usage of AccuACL. As shown in Figure 2(b) of the revised paper, **AccuACL maintains manageable memory requirements compared to AL baselines**, due to its linear complexity concerning the number of unlabeled examples. In short, AccuACL has an average memory usage of 8.3MB, which is **85.5 times smaller** than the full Fisher-based AL, making our algorithm easily deployable in CL environments.
>
> **`See Figure 2(b) of our revised draft.`**
>
> -----

---

> ### Author Response · Authors · 2024-11-25
> **Additional Reponse to Reviewer Fzgq**
>
> Thank you for your response. It is unfortunate that our responses were not satisfactory to you. We would like to further clarify the following points:
>
> `Accuracy Results and State-of-the-Art Comparison:`
> The observed accuracy is influenced by experimental settings, such as buffer size, backbone architecture, and optimization methods, which differ across studies. We are confident that **the accuracy will reach the level of the referenced methods if we use the same experimental setting**. Therefore, we will present the results where AccuACL uses the setting aligned with that in [2] by the end of the rebuttal period.
>
> `Focus of Our Work:`
> Our primary goal is to prevent catastrophic forgetting in the ***active* continual learning (ACL)** environment, not in a typical continual learning (CL) environment. As far as we know, this is the first work that addresses catastrophic forgetting in the ACL problem.
>
> `Space Complexity and Memory Use:`
> $m$ is the size of the memory buffer, and the memory buffer does **not** grow as CL progresses. Some old examples are replaced with new examples. $m$ was fixed to a value from 100 to 5000, depending on the dataset in the paper. In addition, $d$ is the dimensionality of an embedding and was fixed to 512 in the paper. Overall, $m$, $d$, and $K$ are constant throughout the entire CL process.
>
> We are happy to discuss your remaining concerns. Importantly, we will get back to you with additional experimental results using the other setting. Thank you very much.

---

> ### Author Response · Authors · 2024-12-02
> **High Accuracy Results**
>
> To validate the practicality of ACL in high-performing CL settings, we changed only the backbone to a ViT pretrained on ImageNet. As shown in Table A2, the average accuracy has increased significantly. Importantly, AccuACL maintains its effectiveness, achieving performance close to the fully-supervised setting. That is, the absolute accuracy values are heavily influenced by the experiment configuration. These results indicate that **our framework retains its efficacy irrespective of the accuracy value range**. We hope that the main issue you raised has been addressed.
>
> ***Table A2***: Comparison CL and ACL setting for SplitCIFAR10 and SplitCIFAR100 using ER($M$=2000) on a pre-trained ViT.
> | |S-C10| | S-C100| |
> |----|----|----|----|----|
> |         	| $A_{5}$ | $F_{5}$ 	|$A_{10}$   |$F_{10}$ 	|
> | Full    	| 88.7     | 10.3     	|65.2     	| 30.0     	|
> | AccuACL 	| 87.9 	   | 9.4 	    |64.0 	    | 29.6 	    |

---

### Official Review · Reviewer_XnjR · 2024-11-03

**Soundness:** 3
**Presentation:** 4
**Contribution:** 3
**Rating:** 8
**Confidence:** 3

**Summary:**

This paper introduces a novel active learning (AL) approach for active continual learning (ACL) that effectively balances the prevention of catastrophic forgetting with the capacity to quickly learn new tasks. The authors present an accumulated informativeness-based method, termed AccuACL, which extends Fisher information-based AL to a class-incremental setting. By leveraging Fisher information embeddings, AccuACL estimates informativeness through a combination of magnitude and distribution scores, offering a more efficient way to rank data for selection. Experiments across four continual learning (CL) methods on three CL benchmarks show that AccuACL significantly improves CL performance, outperforming traditional AL baselines in terms of both accuracy and forgetting metrics.

**Strengths:**

- The paper addresses active continual learning (ACL), an underexplored area in the intersections of AL and CL.
- In addition to promising experimental results, the paper provides a solid theoretical foundation for the proposed method, as well as an efficient approximation strategy for computing accumulated informativeness scores.
- The authors include analyses of both time and space complexity, demonstrating the feasibility of implementing AccuACL.

**Weaknesses:**

-  Some experiment details like annotation budgets and number of runs are not provided, which are crucial for reproducing the results.
-  Further analysis of the specific data points chosen by each AL method across different CL settings could offer more insight into optimal query selection. Additionally, an examination of annotation budget sensitivity (as many methods peak around round 10 in Figure 2, suggesting diminishing returns) and the effect of task order (since CL can be highly sensitive to task sequencing) would add depth to the findings.

**Questions:**

1. In the main experiments, it’s intriguing to see ACL methods, especially AccuCL and random, outperform training on the full dataset. What annotation budget is used, and how many runs are conducted in this experiment? Also, why is there no standard deviation for the BAIT rows?
2. Figure 2 shows an interesting trend: accuracy and forgetting metrics peak around AL round 10 and then decline for all AL methods. Could you provide an explanation for this phenomenon?

---

> ### Author Response · Authors · 2024-11-23
> **Author's Response to Reviewer XnjR (1/2)**
>
> > W1. Some experiment details like annotation budgets and number of runs are not provided, which are crucial for reproducing the results.
>
> > Q1. ... how many runs are conducted in this experiment? Also, why is there no standard deviation for the BAIT rows?
>
> We appreciate the reviewer bringing this to our attention. We query {1000, 2000, 3000} examples during ten rounds of AL for SplitCIFAR10, SplitCIFAR100, SplitTinyImageNet for each task respectively. Moreover, we have conducted the experiments over three runs with random seeds for each experiment. The values for BAIT lack standard deviation as performance was measured in a single experiment due to high computational costs.
>
> **`See Appendix H of our revised draft.`**
>
> -----
>
> > W2. Further analysis of the specific data points chosen by each AL method across different CL settings could offer more insight into optimal query selection.
>
> We appreciate the reviewer's suggestion to analyze query selection. In response, we conducted additional analysis on SplitCIFAR10, examining data points selected by various AL baselines, as shown in **Figure 6** of the revised draft. While AccuACL does not explicitly utilize specific visual features for sampling, we present these findings to provide insights and allow potential future research.
>
> **`See Figure 6 of our revised draft.`**
>
> -----
>
> > W2. ... examination of annotation budget sensitivity (as many methods peak around round 10 in Figure 2, suggesting diminishing returns) would add depth to the findings.
>
> Thank you for your suggestion. We have conducted additional experiments on SplitCIFAR100 by examining the average accuracy trend according to the number of per-task queried examples (See Table R1 below). Overall, **AccuACL consistently outperforms AL baselines with increasing query numbers**, highlighting its robustness and adaptability.
>
> ***Table R1***: Average accuracy of SplitCIFAR100($M$=500) on ER for different labeling budgets per task. Standard deviation is presented in parentheses.
> | Per-task budget        | 100   | 500   | 1000   | 1500   | 2000   | 2500   | 3000   |
> |:----------|:---------------|:---------------|:----------------|:----------------|:----------------|:----------------|:----------------|
> | Uniform   | **6.2**(0.3)       | 10.8(0.3)      | 10.9(0.3)       | 10.7(0.1)       | 11.0(0.4)       | 11.0(0.3)       | 11.6(0.5)       |
> | Entropy   | 4.8(0.3)       | 7.3(0.7)       | 7.6(0.1)        | 8.0(0.5)        | 8.5(0.3)        | 8.8(0.4)        | 9.6(0.3)        |
> | LeastConf | 4.7(0.6)       | 7.7(0.7)       | 7.4(0.3)        | 7.9(0.2)        | 8.8(0.1)        | 8.9(0.1)        | 9.3(0.4)        |
> | kCenter   | 4.6(0.3)       | 9.0(0.4)       | 9.2(0.2)        | 9.6(0.6)        | 9.7(0.7)        | 10.3(0.5)       | 10.4(0.0)       |
> | BADGE     | 5.6(0.1)       | 9.5(0.7)       | 8.8(0.1)        | 9.1(0.5)        | 9.0(0.0)        | 9.4(0.4)        | 9.6(0.0)        |
> | AccuACL   | 5.5(0.4)       | **12.2**(0.6)      | **14.4**(1.5)       | **15.1**(0.8)       | **14.5**(0.6)       | **15.0**(0.3)       | **13.4**(0.6)       |
>
> **`See Figure 2(a) of our revised draft.`**
>
> -----

---

> ### Author Response · Authors · 2024-11-23
> **Author's Response to Reviewer XnjR (2/2)**
>
> > W2. ... examination of the effect of task order (since CL can be highly sensitive to task sequencing) would add depth to the findings.
>
> We thank the reviewer for suggesting an examination of task order effects. To address this comment, we have conducted additional experiments on SplitCIFAR100 with randomly permuted task orders, where the original dataset is divided into 10 tasks of 10 classes in each sequential order. As shown in Table R2, **AccuACL consistently outperforms the second-best algorithm (Uniform) across all permutations**, demonstrating its robustness and effectiveness regardless of the task order.
>
>
> ***Table R2***: Performance comparison of AccuACL and Uniform across multiple task order permutation on SplitCIFAR100($M$=500), trained with ER. Standard deviation is presented in parentheses.
> |             | Original  |   | Perm. \#1 |   | Perm. \#2  |   | Perm. \#3|   | Perm. \#4  |   |
> |-------------|-----------|-----------|-----------|-----------|-----------|-----------|-----------|-----------|-----------|-----------|
> |             | $A_{10}$  | $F_{10}$  | $A_{10}$  | $F_{10}$  | $A_{10}$  | $F_{10}$  | $A_{10}$  | $F_{10}$  | $A_{10}$  | $F_{10}$  |
> | Uniform     | 10.9(0.4) | 63.7(0.6) | 11.7(0.4) | 62.0(0.3) | 11.1(0.2) | 61.0(1.2) | 11.9(0.4) | 62.5(0.5) | 11.9(0.4) | 61.4(0.5) |
> | **AccuACL** | **14.1**(0.7) | **55.8**(0.9) | **14.7**(1.1) | **52.8**(1.1) | **14.6**(0.5) | **52.4**(0.6) | **15.4**(0.8) | **53.9**(1.0) | **15.8**(0.4) | **51.2**(0.4) |
>
> **`See Appendix E.5 of our revised draft.`**
>
> -----
>
> > Q1. In the main experiments, it's intriguing to see ACL methods, especially AccuCL and random, outperform training on the full dataset.
>
> We appreciate the reviewer's insightful question. The observed performance surpassing the full dataset aligns with analogous findings in core-set selection for CL [a], which enhances performance by selecting high-quality subsets that represent the dataset while avoiding noise or redundancy. Similarly, AccuACL evaluates instance quality in an **unsupervised manner**, constructing high-quality subsets that enable ACL to outperform CL in specific settings. In our experiments, the settings where ACL outperforms CL occur primarily with AccuACL, further highlighting the capability of our algorithm.
>
> **`See Section 5.2 (lines 458-465) of our revised draft.`**
>
> [a] Online Coreset Selection for Rehearsal-based Continual Learning, ICLR, 2022
>
> -----
>
> > Q2. Figure 2 shows an interesting trend: accuracy and forgetting metrics peak around AL round 10 and then decline for all AL methods. Could you provide an explanation for this phenomenon?
>
> We appreciate the reviewer's insightful question. Figure 2 (Figure 5 in the revised draft) examines the effect of varying labeling budgets within a single round. For past (1st-4th) tasks, AL was conducted up to 10 rounds. **Extending beyond 10 rounds for the new (5th) task results in an imbalanced data distribution**, likely causing the observed performance drop. Recognizing that this approach may not be the most effective way to analyze budget sensitivity, we revised Figure 2(a) to explore budget variations across **all** tasks, providing a more balanced and comprehensive analysis.
>
> **`See Figure 2(a) and Appendix E.2 lines(916-917) of our revised draft.`**

---

> ### Author Response · Authors · 2024-11-27
> **Gentle Reminder**
>
> We greatly appreciate your insightful feedback and genuinely hope our response has resolved your concerns. Should any concerns persist, we are willing to engage in discussion with you throughout the remainder of the discussion period.

---

> > ### Comment · Reviewer_XnjR · 2024-11-28
> >
> > Thanks to the authors for clarifying. It addresses my concern, and I'd like to stay positive about supporting the paper's acceptance.

---

> > > ### Author Response · Authors · 2024-11-28
> > >
> > > We deeply appreciate your invaluable feedback and support toward acceptance. Once again, thank you for your insightful and constructive comments.

---

### Official Review · Reviewer_k3jB · 2024-11-03

**Soundness:** 3
**Presentation:** 3
**Contribution:** 3
**Rating:** 8
**Confidence:** 4

**Summary:**

This paper addresses Active Continual Learning (ACL) and particularly the problem of catastrophic forgetting when active learning and continual learning meet. The authors propose propose AccuACL, (Accumulated informativeness-based Active Continual Learning,) which attempts to achieve an optimal balance between learning new knowledge while preserving past knowledge in ACL. They model the accumulated informativeness via the Fisher information matrix,  through the approximation with a small memory buffer, the model parameters, and the unlabeled data pool for the new task. The are theoretical guarantees and extensive experimentation.

**Strengths:**

The novelty of the method is the application of active learning in continual learning. A technique that is applied in active learning is modified to balance learning and prevention of forgetting in the continual setting domain.

Since the proposed solution has a theoretical basis, this basis can be extended to continual learning. The paper includes a theoretical analysis of space and time complexity and also the Fisher Optimality preserving properties.

The proposed method and the intuition behind it are clearly explained in the paper.

The experimentation evaluates the proposed methods from a number of perspectives.

**Weaknesses:**

What are the open problems or future directions related to your work? Adding this to the paper would improve the paper's discussion of broader impact and potential future work. This should be additional to the limitations section in the Appendix.

Typos: Please do a grammar check on your paper. There is a typo in the Abstract “informativeness”

**Questions:**

What are the characteristics of problems where the method works well? Additionally, what are the characteristics of problems where the method does not perform well?

---

> ### Author Response · Authors · 2024-11-23
> **Author's Response to Reviewer k3jB**
>
> > W1. What are the open problems or future directions related to your work? Adding this to the paper would improve the paper's discussion of broader impact and potential future work. This should be additional to the limitations section in the Appendix.
>
> Thank you for highlighting the importance of discussing open problems and future directions. In response, we have added detailed discussions in our revised draft, addressing key avenues such as developing ACL methods for vision-language models (VLMs) [a,b], combining AccuACL with regularization-based CL methods to potentially achieve a rehearsal-free ACL, and designing adaptive ACL methods suitable for realistic scenarios, such as imbalanced CL [c]. We hope that these additions provide a strong foundation for future research.
>
> **`See Appendix I of our revised draft.`**
>
> [a] Active Prompt Learning in Vision Language Models, CVPR, 2024
>
> [b] Boosting continual learning of vision-language models via mixture-of-experts adapters, CVPR, 2024
>
> [c] Online continual learning from imbalanced data, ICML, 2020
>
> -----
>
> > W2. Typos: Please do a grammar check on your paper. There is a typo in the Abstract “informativeness”
>
>
> Thank you for the feedback. We have corrected the typo in the abstract and revised the grammar. We will continue reviewing the draft for any remaining errors.
>
> -----
>
> > Q1. What are the characteristics of problems where the method works well? Additionally, what are the characteristics of problems where the method does not perform well?
>
>
> We appreciate the reviewer for their insightful comment. We suggest that AccuACL performs well when rehearsal-based CL uses memory management strategies that summarize the overall data distribution, enabling accurate Fisher information matrix estimation. While smaller memory buffers pose challenges, our results (e.g., SplitCIFAR100 with $M$=500) show **AccuACL remains effective with as few as 5 examples per class, demonstrating robustness under restrictive conditions.**
>
> -----

---

> ### Author Response · Authors · 2024-11-27
> **Gentle Reminder**
>
> We greatly appreciate your insightful feedback and genuinely hope our response has resolved your concerns. Should any concerns persist, we are willing to engage in discussion with you throughout the remainder of the discussion period.

---

### Author Response · Authors · 2024-11-23
**Response Summary to All Reviewers**

We are extremely grateful for the reviewers' constructive comments and valuable feedback, which have significantly enhanced our paper. Overall, the reviewers have recognized the need for an AL strategy in CL scenarios (all reviewers), strong theoretical background ([k3jB](https://openreview.net/forum?id=mnLmmtW7HO&noteId=tiHphRX72n), [fGgu](https://openreview.net/forum?id=mnLmmtW7HO&noteId=6Jz6jpBM8f), [XnjR](https://openreview.net/forum?id=mnLmmtW7HO&noteId=XJJyiihdzc)), and extensive experimental results ([k3jB](https://openreview.net/forum?id=mnLmmtW7HO&noteId=tiHphRX72n), [fGgu](https://openreview.net/forum?id=mnLmmtW7HO&noteId=6Jz6jpBM8f), [as7J](https://openreview.net/forum?id=mnLmmtW7HO&noteId=dxK0Fju6qn)). Additionally, the reviewers have suggested insightful additional experiments to further establish the efficacy of **AccuACL**. In accordance with the reviewers' feedback, the revised sections of the paper have been highlighted in **blue**. The revisions encompass the followings:

- **Additional Evaluation \& Analysis**
    - Evaluating the performance of **AccuACL** across different labeling budgets (Figure 2(a), Reviewer [XnjR](https://openreview.net/forum?id=mnLmmtW7HO&noteId=XJJyiihdzc), [fGgu](https://openreview.net/forum?id=mnLmmtW7HO&noteId=6Jz6jpBM8f))
    - Evaluating **AccuACL** on *inter-task discrimination-based CL* (Figure 4, Reviewer [Fzgq](https://openreview.net/forum?id=mnLmmtW7HO&noteId=FWIJVdfD2E))
    - Evaluating **AccuACL** on more recent CL method (Table 4, Reviewer [Fzgq](https://openreview.net/forum?id=mnLmmtW7HO&noteId=FWIJVdfD2E))
    - Evaluating the effectiveness of the Fisher information matrix via memory buffer (Table 5, Reviewer [fGgu](https://openreview.net/forum?id=mnLmmtW7HO&noteId=6Jz6jpBM8f))
    - Evaluating the ACL performance with *Learning Accuracy* (Table 6, Reviewer [as7J](https://openreview.net/forum?id=mnLmmtW7HO&noteId=dxK0Fju6qn))
    - Evaluating robustness of **AccuACL** with varying task orders (Table 7, Reviewer [XnjR](https://openreview.net/forum?id=mnLmmtW7HO&noteId=XJJyiihdzc))
    - Adding space complexity analysis of AL baselines (Figure 2(b), Reviewer [Fzgq](https://openreview.net/forum?id=mnLmmtW7HO&noteId=FWIJVdfD2E), [as7J](https://openreview.net/forum?id=mnLmmtW7HO&noteId=dxK0Fju6qn))
- **Enhanced Presentation \& Discussion**
    - Improving Algorithm 1 for readability (Algorithm 1, Reviewer [fGgu](https://openreview.net/forum?id=mnLmmtW7HO&noteId=6Jz6jpBM8f))
    - Improving the limitation and future works (Appendix I, Reviewer [XnjR](https://openreview.net/forum?id=mnLmmtW7HO&noteId=XJJyiihdzc), [fGgu](https://openreview.net/forum?id=mnLmmtW7HO&noteId=6Jz6jpBM8f), [as7J](https://openreview.net/forum?id=mnLmmtW7HO&noteId=dxK0Fju6qn))
    - Improving related works for both AL and CL (lines 98-100, lines 110-112, lines 115-116 Reviewer [Fzgq](https://openreview.net/forum?id=mnLmmtW7HO&noteId=FWIJVdfD2E), [as7J](https://openreview.net/forum?id=mnLmmtW7HO&noteId=dxK0Fju6qn))
    - Providing detailed experimental setup (line 1092-1094, Reviewer [XnjR](https://openreview.net/forum?id=mnLmmtW7HO&noteId=XJJyiihdzc))
    - Providing visualization of query results (Figure 6, Reviewer [XnjR](https://openreview.net/forum?id=mnLmmtW7HO&noteId=XJJyiihdzc))


We believe that the following responses address all the issues and questions raised by the reviewers. Again, we are grateful for the opportunity to address these comments and improve our work.

---

### Comment · Area_Chair_8J5Q · 2024-11-24

Dear Reviewers,

This is a gentle reminder that the authors have submitted their rebuttal, and the discussion period will conclude on November 26th AoE. To ensure a constructive and meaningful discussion, we kindly ask that you review the rebuttal as soon as possible and verify if your questions and comments have been adequately addressed.

We greatly appreciate your time, effort, and thoughtful contributions to this process.

Best regards,
AC

---

### Comment · Area_Chair_8J5Q · 2024-11-27

Dear Reviewers,

We wanted to let you know that the discussion period has been extended to December 2nd. If you haven't had the opportunity yet, we kindly encourage you to read the rebuttal at your earliest convenience and verify whether your questions and comments have been fully addressed.

We sincerely appreciate your time, effort, and thoughtful contributions to this process.

Best,

AC

---

### Meta-Review · Area_Chair_8J5Q · 2024-12-13

**Metareview:**

**Summary**

This work proposes an active learning strategy for continual learning where given a pool of unlabeled data and an annotation budget, the algorithms selects the most relevant samples to annotate for each task increment. It achieves state-of-the-art results compared to other ACL methods.

**Strengths**:
* The framework bridges active learning (AL) with continual learning (CL), proposing an informative sample selection mechanism based on Fisher information.
* The authors provide a solid theoretical justification demonstrating the effectiveness of the sample selection strategy.
* Extensive experiments show significant improvements in accuracy and forgetting metrics over ACL baselines across multiple datasets and continual learning methods.
* The approach introduces efficient approximations for Fisher information, making the method computationally feasible.
* Robustness: Tests with different budgets, task orders, and class-incremental setups demonstrate the adaptability of the method.

**Weaknesses**
* ACL is an artificial problem which might not translate to a real world situation.

Overall, this work is interesting, the algorithm is sound, and it produces state-of-the-art results in ACL. Another point I have taken into consideration is that the field of active continual learning has not progressed much during the recent years and this work could serve as a spark to reignite the field.

**Additional Comments On Reviewer Discussion:**

Most reviewers have identified it as valuable and lean towards acceptance (k3jB, fGgu, XnjR, as7J). While Fzgq is the only reviewer leaning towards rejection, I believe the reviewer is judging this work as a continual learning algorithm rather than an active continual learning one and thus I believe this work is still worthy of acceptance at ICLR.

---

### Decision · Program_Chairs · 2025-01-22

Accept (Poster)